# A nutrient-responsive hormonal circuit mediates an inter-tissue program regulating metabolic homeostasis in adult *Drosophila*

Takashi Koyama [1,4], Selim Terhzaz [2,3,4], Muhammad T. Naseem [1], Stanislav Nagy[1], Kim Rewitz [1], Julian A. T. Dow [2], Shireen A. Davies[2] & Kenneth V. Halberg [1✉]

Animals maintain metabolic homeostasis by modulating the activity of specialized organs that adjust internal metabolism to external conditions. However, the hormonal signals coordinating these functions are incompletely characterized. Here we show that six neurosecretory cells in the *Drosophila* central nervous system respond to circulating nutrient levels by releasing Capa hormones, homologs of mammalian neuromedin U, which activate the Capa receptor (CapaR) in peripheral tissues to control energy homeostasis. Loss of Capa/CapaR signaling causes intestinal hypomotility and impaired nutrient absorption, which gradually deplete internal nutrient stores and reduce organismal lifespan. Conversely, increased Capa/CapaR activity increases fluid and waste excretion. Furthermore, Capa/CapaR inhibits the release of glucagon-like adipokinetic hormone from the corpora cardiaca, which restricts energy mobilization from adipose tissue to avoid harmful hyperglycemia. Our results suggest that the Capa/CapaR circuit occupies a central node in a homeostatic program that facilitates the digestion and absorption of nutrients and regulates systemic energy balance.

[1] Section for Cell and Neurobiology, Department of Biology, University of Copenhagen, Universitetsparken 15, Copenhagen, Denmark. [2] Institute of Molecular, Cell and Systems Biology, College of Medical, Veterinary and Life Sciences, University of Glasgow, Glasgow, UK. [3] MRC-University of Glasgow Centre for Virus Research, Glasgow, UK. [4]These authors contributed equally: Takashi Koyama, Selim Terhzaz. ✉email: kahalberg@bio.ku.dk

The ability to maintain metabolic homeostasis in response to environmental challenges is a fundamental prerequisite for animal life. Organisms must ensure a balanced equilibrium between nutrient intake, storage and expenditure, which is governed by complex interorgan communication networks that coordinate the action of specialized organs[1,2]. To implement the correct homeostatic program, animals must translate their metabolic needs into physiological responses that adjust their uptake and utilization of resources accordingly. In mammals, the hypothalamus acts as a central command center for nutrient sensing, as it contains neuronal populations that are activated by changes in extracellular sugar concentration and releases key hormones that initiate compensatory organ activities[3,4]. Similarly, in the fruit fly *Drosophila melanogaster* discrete populations of neurosecretory cells function as nutrient sensors, which upon activation secrete neurohormones that modulate food intake, energy mobilization, gut peristalsis or renal secretion[5–7]. Thus, both mammals and flies regulate organ activities in response to internal state, and in many instances accomplish this regulation by similar mechanisms. However, the hormonal factors involved in this coordination, as well as the mechanisms by which the signals are integrated by different organs are incompletely characterized. Understanding these processes is of vital importance as failure in these systems often results in fatal consequences for the organism.

Our knowledge of Capa/CapaR signaling in insects have largely been restricted to adult animals, in which the Capa peptides are released from subsets of neurosecretory cells in the central nervous system (CNS), to activate the Capa receptor (CapaR) on target tissues, including the renal tubules, heart and hyperneural muscles, to control diuretic and myotropic functions[8]. In *Drosophila*, the *Capa* gene encodes a preprohormone that is processed to produce four different peptides, Capa-periviscerokinin-1 and −2 (Capa-PVK-1 and −2), Capa-pyrokinin-1 (Capa-PK-1) and Capa precursor peptide B (CPPB)[9]. The Capa precursor is differentially processed within different subsets of neurosecretory cells, with a truncated form of Capa-PK-1 released from the subesophageal ganglion (SEG) neurons, while both Capa-PVKs, -PK and -CPPB are secreted from the ventroabdominal (Va) neurons[10]. The Capa peptides are members of the PRXamide peptide family, which is evolutionarily and functionally related to neuromedin U (NmU) signaling in vertebrates[11,12]. In mammals, NmU signaling coordinates key physiological processes including visceral-muscle contractions, gastric acid secretion and insulin release as well as feeding and energy homeostasis, and is therefore an attractive therapeutic target for treating obesity[13].

Here, we report that Capa-PVK signaling (hereafter just Capa) plays a vital role in regulating adult metabolic homeostasis in response to environmental conditions. Detailed mapping of CapaR expression identified the visceral musculature, enteroendocrine cells (EEs) and neuroendocrine cells of the corpora cardiaca (CC)—producing the glucagon analog adipokinetic hormone (AKH)—as targets of Capa action. Loss of Capa/CapaR signaling reduces intestinal contractility, gut regionalization and nutrient absorption, which result in systemic metabolic defects characterized by pronounced hypoglycemia and lipodystrophy. These metabolic effects cause a gradual loss of muscle function due to dysregulated $Ca^{2+}$ homeostasis in skeletal muscles, which impairs feeding behavior and causes premature mortality. We further show that the Capa Va neurons, but not the SEG neurons, secrete Capa peptides in response to nutrient availability to repress AKH release to restrict energy mobilization from the fat body during nutrient-replete states. We propose that the Capa Va neurons operate as post-ingestive nutrient-sensors that coordinate a CNS-gut-renal-fat body signaling module to activate two separate pathways: one to enhance intestinal and renal activities to promote energy and fluid homeostasis, and another to inhibit AKH-mediated energy mobilization to prevent hyperglycemia. Our work thus uncovers an adult-specific inter-tissue program that is essential to maintain osmotic and metabolic homeostasis in *Drosophila*—a program that shows remarkable functional similarity with NmU signaling in mammals.

## Results

**Capa/CapaR signaling is essential for adult fly survival.** To gain insight into the physiological actions of Capa/CapaR signaling, we used the binary GAL4/UAS system to silence *CapaR* gene expression using RNAi (Supplementary Fig. 1a, b). Whereas juvenile male growth and developmental time were unaffected (Supplementary Fig 1c, d), knockdown of *CapaR* expression using *CapaR^RNAi* driven by *CapaR-GAL4*[11] resulted in strong mortality shortly after adult emergence. Knockdown efficiency and the penetrance of phenotypic effects scaled with temperature, at least in part, due to the thermal flexibility of the GAL4/UAS system[14] (Fig. 1a, b; median survival time was 10, 3, and 1 days at 18 °C, 25 °C, and 30 °C, respectively). RNAi specificity was confirmed with two additional independent *CapaR^RNAi* lines that produced similar phenotypes (Supplementary Fig. 1f). To unambiguously demonstrate a post-developmental role of Capa/CapaR signaling in sustaining adult survival, we adopted an alternative genetic strategy in which we designed a UAS-inducible tissue-specific CRISPR/Cas9 construct for *CapaR* (Supplementary Fig. 1a, e), which we spatio-temporally restricted to the adult stage using the TARGET system[15]. Consistent with the phenotypes observed following *CapaR* knockdown, we confirmed that adult-specific *CapaR* knockout caused complete fly mortality within 10 days following transfer to the restrictive temperature. Crucially, the lethality caused by *CapaR* deletion was almost fully rescued in flies additionally carrying a *UAS-CapaR* transgene immune to CRISPR-induced mutation (Fig. 1b).

To identify the tissue-specific actions underlying this acute mortality, we selectively downregulated expression of *CapaR* in different tissues. Given that Capa peptides are known to modulate renal function in *Drosophila* and other insects[16], we knocked down *CapaR* only in the Malpighian tubules (MTs), where the receptor is abundantly expressed (Supplementary Fig. 2a–c), and assessed adult survival. Silencing *CapaR* expression using the *Uro-GAL4* driver, which exclusively targets the principal cells of the MTs[17], did not cause any significant mortality (Supplementary Fig. 1g, h), suggesting that the lethality phenotype is likely uncoupled from Capa-dependent regulation of renal function. We therefore assayed the phenotypic effects of *CapaR* knockdown in other major tissues, including the muscles (*how-GAL4*), neurons (panneuronal *elav-GAL4* or mushroom-body-specific *201Y-GAL4*), fat body (*c564-GAL4 or Cg-GAL4*), and heart (*Tinman-GAL4 or Hand-GAL4*). These data revealed that only *how-GAL4*, which expresses GAL4 in the somatic and visceral musculature (Supplementary Fig. 1i, j)[18], could phenocopy the mortality observed with the *CapaR-GAL4* driver (Fig. 1c). Importantly, this mortality phenotype was recapitulated by precise targeting of the somatic CRISPR/Cas9 construct to adult muscles, which similar to global knockout (Fig. 1b), caused the majority of flies to die within a ten-day period. Similarly, this mortality was completely rescued to levels approaching that of controls in flies additionally expressing a transgenic *CapaR* receptor in the muscles (Fig. 1d). Taken together, these findings link Capa/CapaR signaling to targets outside its well-established role in regulating renal function[11] and suggest that Capa peptides coordinate additional processes in the musculature that are essential to sustain adult survival.

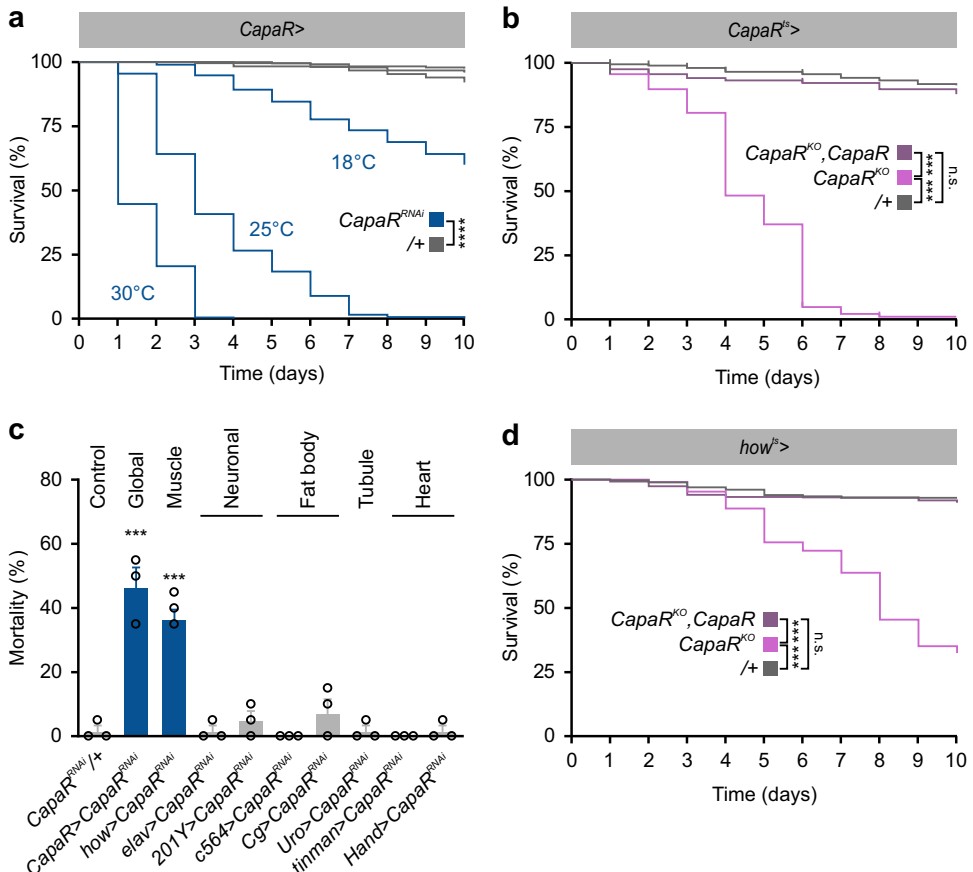

**Fig. 1 Loss of *CapaR* in muscles decreases adult survival. a** Global knockdown of *CapaR* using the in-house generated doubly homozygous 2xRNAi line (*CapaR > CapaR^RNAi*) results in strong mortality shortly after adult emergence. Lethality correlates with increased temperature due to the thermal sensitivity of the GAL4/UAS-system. Median survival for male flies was 10, 3, and 1 days at 18 °C ($n = 213$), 25 °C ($n = 293$) and 30 °C ($n = 232$), respectively (two-sided log-rank test; $P < 0.0001$). The experiment was repeated three times with the same results. **b** CRISPR-mediated adult-specific *CapaR* knockout using the *tub-GAL80^ts* construct (*CapaR^ts > CapaR^KO*; $n = 498$) induces significant mortality in male flies compared to controls (*CapaR^ts > +*; $n = 436$; two-sided log-rank test; $P < 0.001$). This lethality was fully rescued in flies additionally carrying a *UAS-CapaR* transgene (*CapaR^ts > CapaR^KO,CapaR*; $n = 451$). The experiment was repeated multiple times with the same results. **c** Knockdown of *CapaR* (*CapaR^RNAi*) in whole animals (*CapaR>*) or exclusively in muscles (*how>*), but not in neurons (*elav>* or *201Y>*), fat body (*c564>* or *Cg>*), Malpighian tubules (*Uro>*) or heart (*tinman>* or *hand>*), cause significant male fly mortality (mean ± SEM; one-way-ANOVA; $n = 3$; $P < 0.001$). The experiment was repeated twice with the same results. **d** Muscle-specific CapaR knockout (*how^ts > CapaR^KO*) recapitulates the lethality phenotype, with knockout flies ($n = 793$) showing significantly higher mortality compared to control (*how^ts>+*; two-sided log-rank test; $n = 723$; $P < 0.001$), and survival is completely rescued in flies co-expressing a receptor transgene (*how^ts > CapaR^KO,CapaR*; $n = 600$). The experiment was repeated multiple times with the same results.

**Capa/CapaR signaling targets several tissues**. To identify the specific cellular targets of Capa/CapaR signaling, we analyzed CapaR expression using a combination of genetic and molecular reporters. As expected, *CapaR-GAL4* was found to induce strong GFP fluorescence in both the anterior and posterior MTs (Supplementary Fig. 2a–c); however, we additionally detected prominent and consistent reporter activity in the thoracic skeletal muscles, brain as well as circular visceral muscles confined to the proventriculus, midgut and rectal regions of the intestine (Fig. 2a; Supplementary Fig. 2a). The accurate reporting of endogenous *CapaR* localization was verified using a custom-synthesized CapaR antibody[11], which showed substantial overlap between GFP fluorescence and anti-CapaR immunoreactivity (Fig. 2a). These findings were further corroborated using fluorescently tagged Capa-1 peptide (Capa-1-F) in combination with a recently developed ligand-receptor binding assay[16]; fluorophore labeling did not affect the biological efficacy of the peptide (Supplementary Fig. 2d–g). Using this approach, we observed specific and displaceable Capa-1-F binding to *CapaR-GAL4*-driven GFP-positive visceral muscles and EEs (Fig. 2b, c) of the adult

digestive tract, which is consistent with single-cell transcriptomic data reported by Flygut-seq[19] showing that *CapaR* expression is almost entirely restricted to these two cell types (Supplementary Fig. 2h). The EE identity of the CapaR expressing cells was verified by colocalization of the homeodomain protein Prospero[20], and these cells were further shown to co-express the gut hormones Allatostatin C (AstC), CCHamide1 (CCHa1) and Tachykinin (TK), but not Neuropeptide F (NPF) or Diuretic Hormone 31 (DH31) (Supplementary Fig. 2i).

Finally, to functionally validate our receptor mapping data, we expressed an aequorin-based bioluminescent $Ca^{2+}$-sensor using *CapaR-GAL4* to detect robust changes in cytosolic $[Ca^{2+}]_i$ following Capa stimulation of acutely dissected tissues (Fig. 2d)[21]. Consistent with previous findings, Capa-1 induced a stereotypic, biphasic rise in $[Ca^{2+}]_i$, comprising a rapid primary peak followed by a slower secondary rise in the renal tubules (Supplementary Fig. 2j)[11]. Yet more strikingly, we also observed a prominent cytosolic calcium response in immuno-positive gut regions such as the proventriculus, midgut and hindgut/rectum, as well as in the brain (Fig. 2e–h), and to a lesser extent in the legs

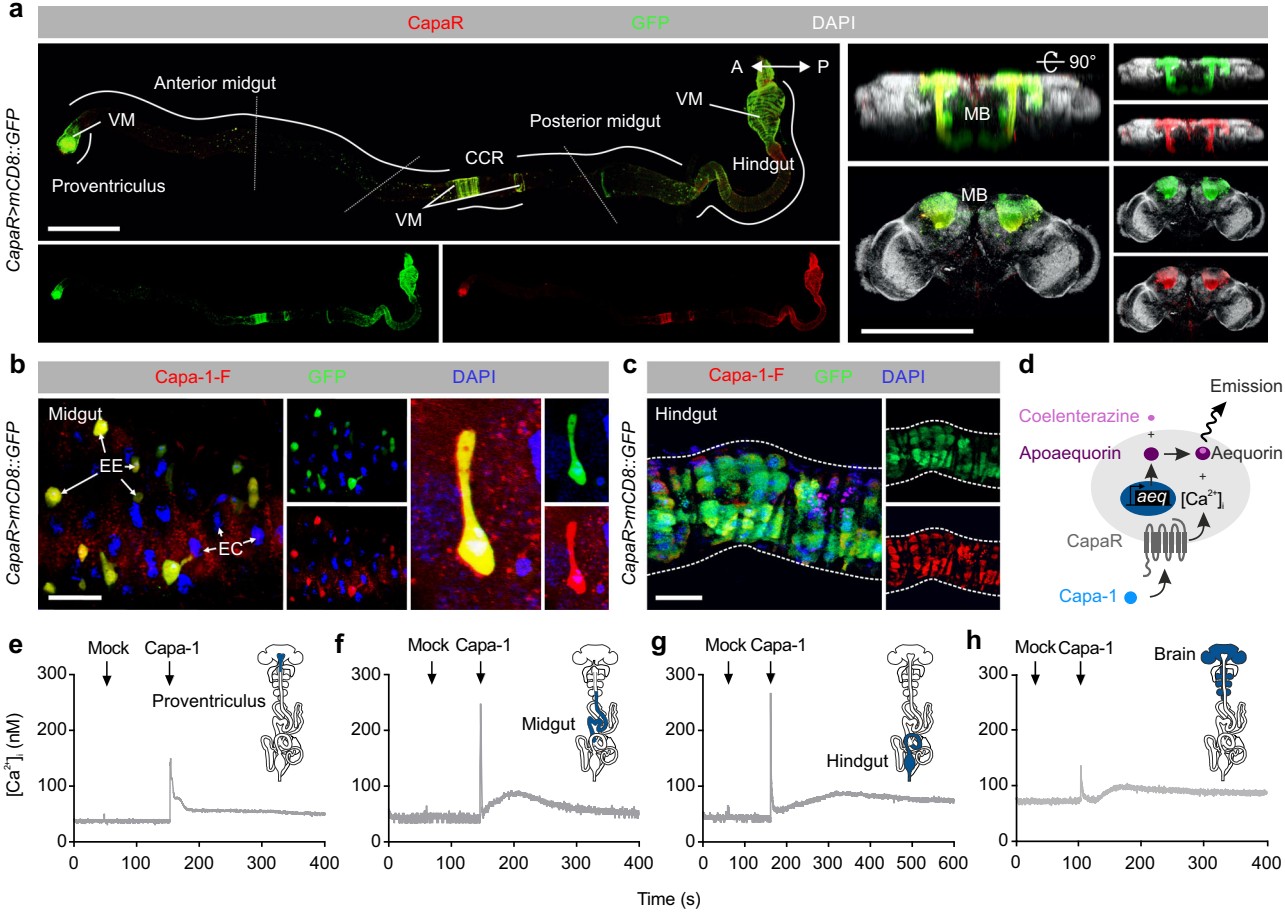

**Fig. 2 Global mapping of CapaR expression. a** Co-immunolabeling of the CNS and the intestine from *CapaR > mCD8::GFP* flies with a custom generated CapaR antibody consistently showed an overlap between GFP (green) and CapaR (red) immunoreactivity in the mushroom body (MB) and neuronal somata of the CNS, as well as in the visceral muscles (VM) and enteroendocrine cells (EEs) restricted to distinct portions of the gut. Image was acquired using "Tile scan" in the ZEN software (Zeiss, Oberkochen, DE) with boundaries between tiles marked by dashed line. EC, enterocytes; CCR, copper-cell region. Scale bars = 200 μm. **b–c** Application of fluorophore-labeled Capa-1 (Capa-1-F, red) on the intestine shows specific and displaceable binding to *CapaR > GFP* (green) positive enteroendocrine cells (scale bar = 20 μm) and visceral muscles (scale bar = 40 μm). **d** Principle of in vivo aequorin luminescence-based functional calcium assay in *Drosophila* tissues. **e–h** Real-time changes in intracellular $Ca^{2+}$ concentration of tissues acutely dissected from flies expressing a *UAS-apoaequorin* transgene (*CapaR > aeq::GFP*). Cytosolic $[Ca^{2+}]_i$ levels (nM) in **e** proventriculus, **f** midgut, **g** hindgut and **h** brain upon Capa-1 stimulation ($10^{-7}$ M).

(Supplementary Fig. 2k); by contrast, we were unable to detect any $Ca^{2+}$ response in the gonads or salivary glands (Supplementary Fig. 2l, m). Taken together, our combined approaches provide a comprehensive overview of CapaR expression in adult *Drosophila*, which unmasks several target tissues of Capa neuropeptides including the skeletal and visceral muscles.

**Four pairs of Capa neurons mediate CNS to peripheral organ communication.** To determine the neurons responsible for the Capa-mediated effects on adult viability, and to enable their subsequent genetic manipulation, we characterized a GAL4 driver line, *Trp-GAL4*, that when crossed to *mCD8::GFP* produced almost complete overlap in immunoreactivity with an anti-Capa precursor antibody[9] in the adult CNS, hereby demonstrating unparalleled specificity and potency to the *Capa*+ neurons (Fig. 3). Indeed, knocking down *Capa* expression, or completely ablating the Capa-producing neurons via overexpression of the proapoptotic gene *rpr* recapitulated the mortality phenotype of *CapaR* manipulations (Supplementary Fig. 1k); in contrast to *CapaR* depletion, silencing or ablation of the Capa+ neurons also induced lethality during larval development, perhaps due to the impairment of additional *Capa* products released by the Va

neurons[10,22]. Performing a detailed neuroanatomical analysis using *Trp-GAL4*, we found consistent GFP expression in one pair of neuroendocrine cells in the SEG, whose dendritic projections innervate the AKH-producing cells (APCs) and extend along the esophagus to innervate the proventriculus (Fig. 3a–c; Supplementary Fig. 3a, c). In addition, we observed reporter activity in the three pairs of Va neurons; the two anterior pairs send axons to a neurohemal plexus in the dorsal neural sheath, while the posterior-most pair joins the median abdominal nerve (MAN; Fig. 3d; Supplementary Fig. 3b), from which their axons project to form neurohemal release sites. This neuroarchitecture strongly suggests that Capa peptides are released into circulation local to their target tissues. The empirical basis underpinning such a model furthermore includes the previously observed fusion of immunogold-labeled Capa+ vesicles released from these sites[23], as well as direct mass spectrometric profiling identifying Capa hormones as the main release products of these neurohemal organs[10,22]. Although direct innervation of the intestine by neurons exiting the CNS through the MAN or the presence of Capa+ enteric neurons would offer an attractive model for Capa-mediated changes in intestinal physiology, we were unable to observe such connections. In sum, our data reveals neuronal

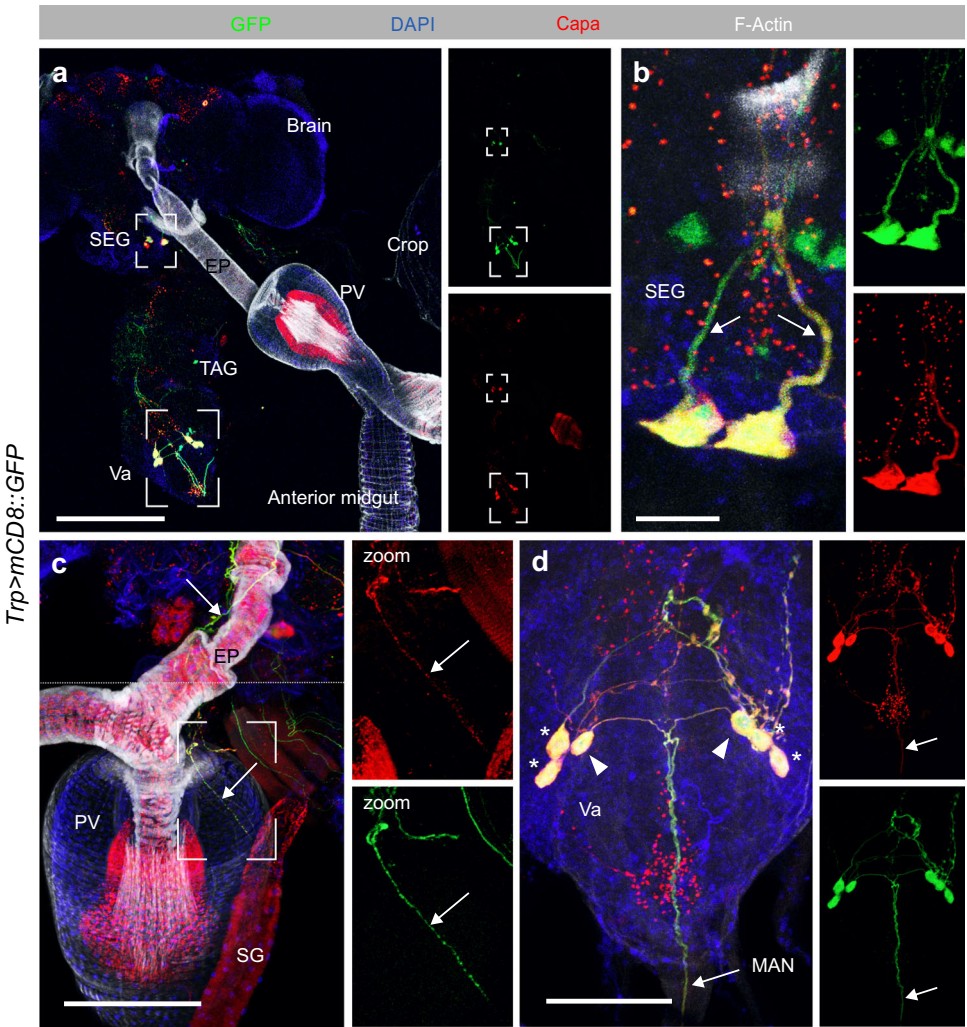

**Fig. 3 Neuroanatomy of Capa-producing neurons. a–d** The CNS and associated tissues showing *Trp > mCD8::GFP* (green) counterstained with an anti-Capa antibody (red), F-Actin (white) and DAPI (blue) in maximum projected z-stack confocal images. The experiment was repeated multiple times with the same results. **a** GFP expression and anti-Capa immunoreactivity co-localize in four pairs of Capa+ neurons; one pair in the subesophageal ganglion (SEG) and three pairs in the thoracicoabdominal ganglion (TAG). Scale bar, 200 μm. **b** The SEG neurons extend dorsally. Scale bar, 20 μm. **c** The SEG neurons also extend along the esophagus (EP) to innervate the proventriculus (PV). Scale bar, 50 μm. Image was acquired using "Tile scan" in the ZEN software (Zeiss, Oberkochen, DE) with the boundary between tiles marked by dashed line. **d** The two anterior pairs (*) of ventroabdominal (Va) neurons send axons to the dorsal neural sheath, while the posterior pair (triangles) projects into the median abdominal nerve (MAN) and form neurohemal release sites. Scale bar, 50 μm.

innervation of the CC and proventriculus by the SEG neurons, and the systemic release of Capa peptides into circulation by the Va neurons to activate peripheral target tissues.

**Capa/CapaR signaling regulates gut motility and waste excretion**. The expression of CapaR in visceral muscles confined to portions of the adult intestine containing valves and sphincters (proventriculus, posterior midgut and rectal pad; Fig. 2a), suggests modulation of intestinal transit by Capa-dependent control of smooth muscle cell activity. Such a role is consistent with the known functions of mammalian Neuromedin U receptor 2 (NmU-R2, a functional homolog of CapaR[11]), which has been shown to promote gastro-intestinal motility in mammals[24]. Using an ex vivo gut-contraction assay[16], we quantified the spontaneous contraction frequency of the intestine from *w1118* flies (the white null background was used for all our transgenics and is therefore the most suitable genetic background for these experiments) before and after Capa-1 or Capa-2 application. We found that both Capa peptides—at a concentration previously determined to cause maximum receptor occupation[11]—

stimulate gut contraction frequency approximately 2-3-fold compared to the artificial hemolymph control saline (Fig. 4a, c). Conversely, the Capa-induced activation of gut motility was abolished in flies in which we knocked down *CapaR* in visceral muscles, indicating that the increase in gut-contraction frequency following Capa-1/-2 application depends on CapaR function in intestinal muscle cells (Fig. 4b, d). CapaR is a Gq-protein coupled receptor in which receptor activation results in elevation of intracellular $Ca^{2+}$-levels. We therefore used the genetically encoded $Ca^{2+}$-indicator *UAS-GCaMP6s*[25] to directly visualize $Ca^{2+}$-dynamics in CapaR-expressing circular muscle cells during Capa-1 stimulation (Fig. 4e). These data show that Capa-1 induces prominent intracellular $Ca^{2+}$-transients in muscle cells supporting the notion that Capa/CapaR stimulates visceral muscle activity.

To explore the physiological significance of this effect, we next asked if the observed impairment of gut motility caused by reduced Capa/CapaR signaling affects intestinal transit time in vivo. In flies exposed to food supplemented with the pH-sensitive dye Bromophenol blue (BPB), we observed a significant delay in gut transit in

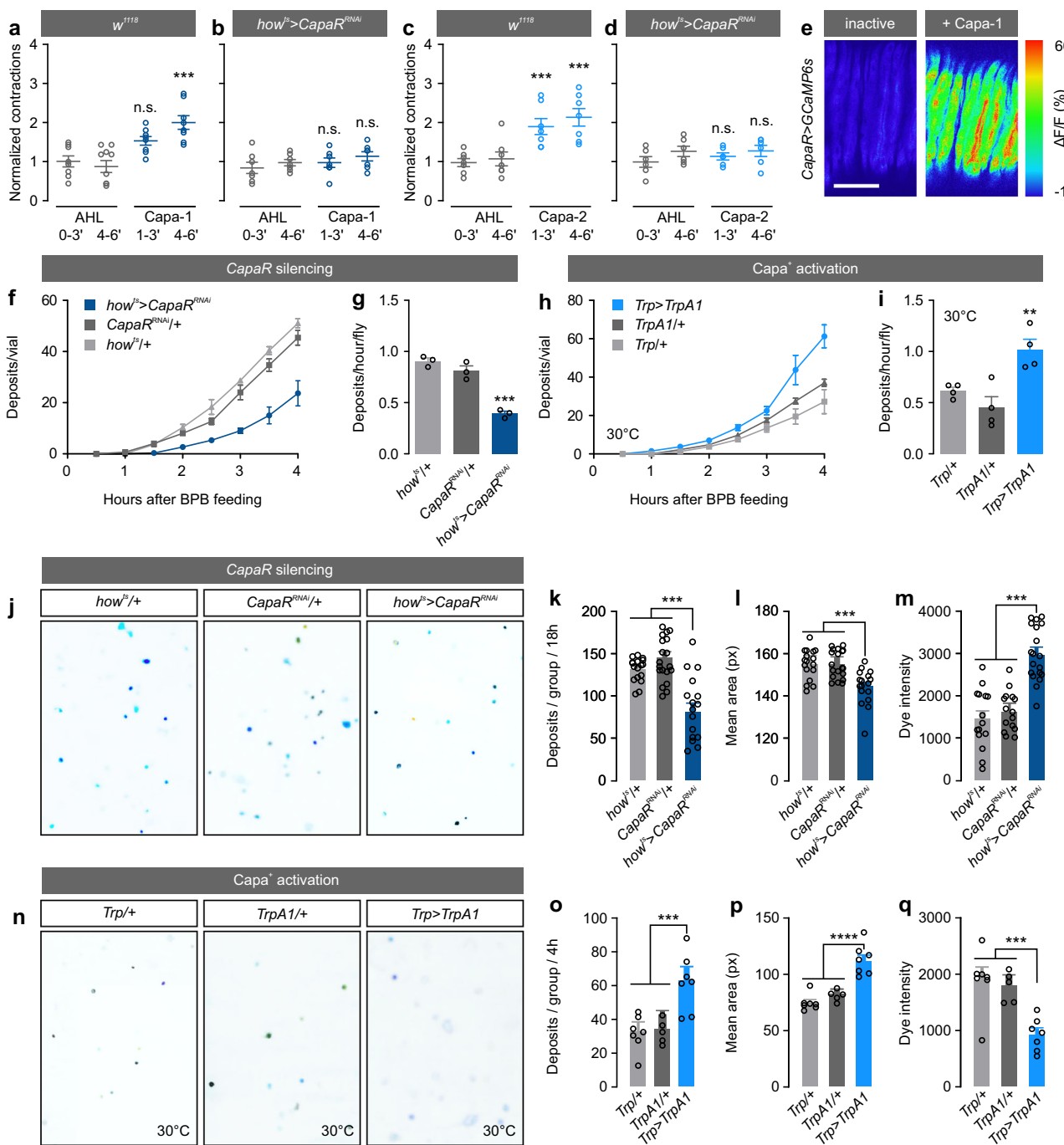

**Fig. 4 Capa modulates intestinal contractions and excretory physiology. a–d** Spontaneous gut contractions from $w^{1118}$ and muscle-specific *CapaR* knockdown flies (*how*$^{ts}$ > *CapaR*$^{RNAi}$) in response to artificial hemolymph (AHL) and AHL containing **a, b** Capa-1 or **c, d** Capa-2 at a concentration of $10^{-7}$ M. Gray values indicate exposures to AHL alone, whereas colored values refer to exposures to AHL + Capa-1 or Capa-2. ΔContractions were calculated by normalizing the number of contractions during Capa-1/-2 application over those in the control (one-way-ANOVA; **a** $n = 8$, $P < 0.0003$; **b** $n = 7$, $P > 0.05$; **c** $n = 7$, $P < 0.001$; **d** $n = 6$, $P > 0.05$). **e** Inactive and activated circular visceral muscles in guts from flies expressing the GCaMP6s calcium sensor (*CapaR* > *GCaMP6s*) following Capa-1 stimulation. The experiment was repeated three times with corresponding results. Scale bar = 100 μm. **f** Accumulated number of deposits and **g** the number of deposits/hour/fly in muscle-specific *CapaR* silenced flies (*how*$^{ts}$ > *CapaR*$^{RNAi}$) compared to parental controls (mean ± SEM; one-way-ANOVA; $P < 0.001$; $n = 3$ vials with 20 flies per vial). **h** Accumulated number of deposits, **i** the number of deposits/hour/ fly in flies with artificial activation of the Capa$^+$ neurons by ectopic expression of the heat-sensitive TrpA1 channel (*Trp* > *TrpA1*) compared to control flies at 30 °C (mean ± SEM; one-way-ANOVA; $P = 0.0033$; $n = 3$ biologically independent replicates with 20 flies per replicate). **j, n** Representative fecal output profiles from the different genetic manipulations fed on BPB-labeled food. Muscle-specific silencing of Capa/CapaR signaling (*how*$^{ts}$ > *CapaR*$^{RNAi}$) results in **k** fewer, **l** smaller, and **m** more concentrated BPB-labeled deposits compared to the parental controls (mean ± SEM; one-way-ANOVA; **k** $n = 17$, $P < 0.001$; **l** $n = 17$, $P = 0.003$, **m** $n = 15$, $P < 0.001$; n-numbers refer to biologically independent replicates with 10 flies per replicate). **o** Conversely, overactivation of Capa-producing neurons (*Trp* > *TrpA1*) induced excretion of more ($P = 0.0006$), **p** larger ($P < 0.0001$), and **q** lighter deposits ($P < 0.0006$) compared to flies carrying each transgene alone (mean ± SEM; one-way-ANOVA; $n = 5$ biologically independent replicates with 10 flies per replicate).

muscle-specific knockdown animals compared to parental controls, as evidenced by the delayed appearance and significant decrease in BPB-labeled waste deposits produced over time (Fig. 4f, g). In contrast, genetic overactivation of the Capa-producing neurons using the heat-activated transient receptor potential A1 cation channel TRPA1 using *Trp-GAL4* led to a significantly shorter intestinal transit time when incubated at 30 °C (Fig. 4h, i). Thus, Capa-1 and -2 hormones signal to the gut to modulate intestinal motility and are necessary and sufficient for the control of gut transit in adult *Drosophila*.

To further explore the functional significance of *CapaR* silencing on gut physiology in vivo, we adopted an approach that provides integrated readouts of intestinal and renal activities based on a quantitative and semi-automated analysis of fly excreta[26,27]. These experiments showed that muscle-specific *CapaR* knockdown significantly affects defecation behavior compared to the parental controls, as these flies produced fewer, smaller, and more concentrated excreta (Fig. 4j–m). These changes in fecal output are likely caused by prolonged contact with intestinal contents associated with gut hypomotility. Consistent with this model, artificial over-activation of the Capa neurons led to the production of lighter (less concentrated), larger and more abundant deposits compared to flies carrying either transgene alone, presumably due to the combined effects of intestinal hypermotility and increased diuresis (Fig. 4n–q). Interestingly, limiting the time available for fluid reabsorption in the gut has a larger impact on systemic fluid balance than stimulating renal secretion, as adult-specific expression of a membrane-tethered version of the diuretic hormone DH44—which allows cell-autonomous activation of its cognate receptor and thus constitutive activation of tubule secretion (Supplementary Fig. 4a, b)[28]—failed to fully recapitulate this diarrhea-like phenotype. Instead, the deposits produced by these flies were more numerous, but not larger or more dilute than those of controls consistent with the role of the rectum in regulating the final fluid content of excreta[29] (Supplementary Fig. 4c). Conversely, flies in which we knocked down *CapaR* expression in the renal tubules produced fewer deposits, albeit of similar size and dye intensity relative to controls (Supplementary Fig. 4d). Together, these results show that Capa peptides direct the actions of the CNS-gut axis by modulating intestinal contractility, which is important not only for appropriate gut transit, but also directly impacts the homeostatic regulation of water balance by affecting fluid content of excreta.

**Capa/CapaR is necessary to maintain systemic metabolic homeostasis**. Next, we asked if these changes in gut physiology affect the metabolic status of the fly. Indeed, in mammals gut hypomotility has been shown to compromise both intestinal acid-base balance, nutrient absorption and organismal energy status[30]. Consistent with these observations, we observed frequent gut distension as well as loss of acidity in the copper-cell region (CCR) of midguts from Capa/CapaR deficient flies (Fig. 5a); a region functionally analogous to the mammalian stomach[31,32]. We therefore tested if these gut defects affect nutrient breakdown and absorption by the gut, by analyzing the elemental composition of deposits from flies lacking *CapaR* function in the visceral muscles using scanning electron microscopy coupled with wave-dispersive X-ray analysis. We detected significantly higher levels of carbon and nitrogen (the main elemental components of sugars, amino acids and lipids) in the excreta of *CapaR* knockdown flies compared to the control (Fig. 5b, c), suggesting a potential disruption of digestive and/or absorptive functions of the gut. To directly test this, we quantified triacylglyceride (TAG) and glucose levels in the excreta from muscle-specific *CapaR*

knockout flies, which revealed that their deposits contained significantly higher levels of residual undigested nutrients compared to controls (Fig. 5d). Intriguingly, these effects were phenocopied by acute knockdown of an essential subunit of the Vacuolar H$^+$-ATPase (*Vha55*)—a membrane transporter critical for acid generation in the CCR[33]—specifically in the midgut acidic region using *Lab-GAL4*[31,32], suggesting that the low pH of the CCR is necessary for optimal nutrient digestion and absorption, as well as long-term survival in adult flies[34] (Supplementary Fig. 5a–d). Importantly, the gut defects observed in *CapaR* knockout animals, including the loss of CCR acidity, were fully rescued by co-expressing a *CapaR* construct (Fig. 5a, d). To further test this model, we exposed muscle-specific *CapaR* knockout flies to different nutrient concentrations to assess the impact of nutrient density on energy uptake and animal viability. These data showed that higher nutrient concentrations significantly prolonged fly survival in a dose-dependent manner, while both control and rescue animals showed no change in mortality (Fig. 5e). These results suggest that *CapaR* elimination in intestinal muscle cells compromises the regional specialization as well as the digestive and absorptive functions of the gut, which is necessary to sustain animal survival.

Reduced intestinal nutrient absorption is likely to alter the nutritional status of the fly and Capa/CapaR deficient animals may therefore show compensatory changes in feeding behavior. To test this hypothesis, we quantified the duration of food contact as an indirect measure of food intake, in addition to directly measuring food consumption. Surprisingly, these data revealed that *CapaR* knockdown flies spent significantly less time in contact with food (Fig. 5f) and that *CapaR* knockout animals reduced their overall dietary intake; an effect that could be reversed by overexpression of *CapaR* in muscles (Fig. 5g). We also observed similar effects upon targeting the Capa$^+$ neurons by *Capa* knockdown (Fig. 5h), indicating that Capa/CapaR signaling may modulate feeding behavior either directly or indirectly in adult flies. This idea is consistent with the role of NmU signaling in regulating food consumption in rodent models[35,36]. The combined effects of reduced energy intake and impaired intestinal absorption strongly suggest that these flies show severe metabolic defects. We therefore assessed the internal nutritional status of these flies, by quantifying internal carbohydrate levels (glucose and trehalose) as well as whole-body energy stores (glycogen and TAG) at distinct points in time following muscle-specific *CapaR* elimination. As expected, *CapaR* knockout animals showed a gradual decrease in internal energy stores, due to increasingly poor feeding and nutrient absorption, as revealed by their progressive hypoglycemic and lipodystrophic phenotypes compared to both control and rescue flies (Fig. 5i–l). Altogether, our data suggest that loss of Capa/CapaR signaling in visceral muscles impairs food intake and cause defects in the digestive and absorptive functions of the gut, which critically impact the internal energy status and lifespan of adult flies.

**Impaired Capa/CapaR signaling reduces muscle performance and locomotion**. In flies, as well as mammals, changes in systemic energy balance induces the release of hormonal factors that modulate behavioral programs[37–39]. For example, central administration of NmU peptides increases gross-locomotor activity in rodents[36]. We therefore asked whether Capa/CapaR impairment induces similar responses in locomotor activity, by examining baseline locomotion of adult flies using the *Drosophila* Activity Monitor System. These data revealed that muscle-specific *CapaR* elimination induced a gradual decrease in adult locomotion over the course of a 4-day experimental period, as evidenced by the significant reductions in both morning and evening

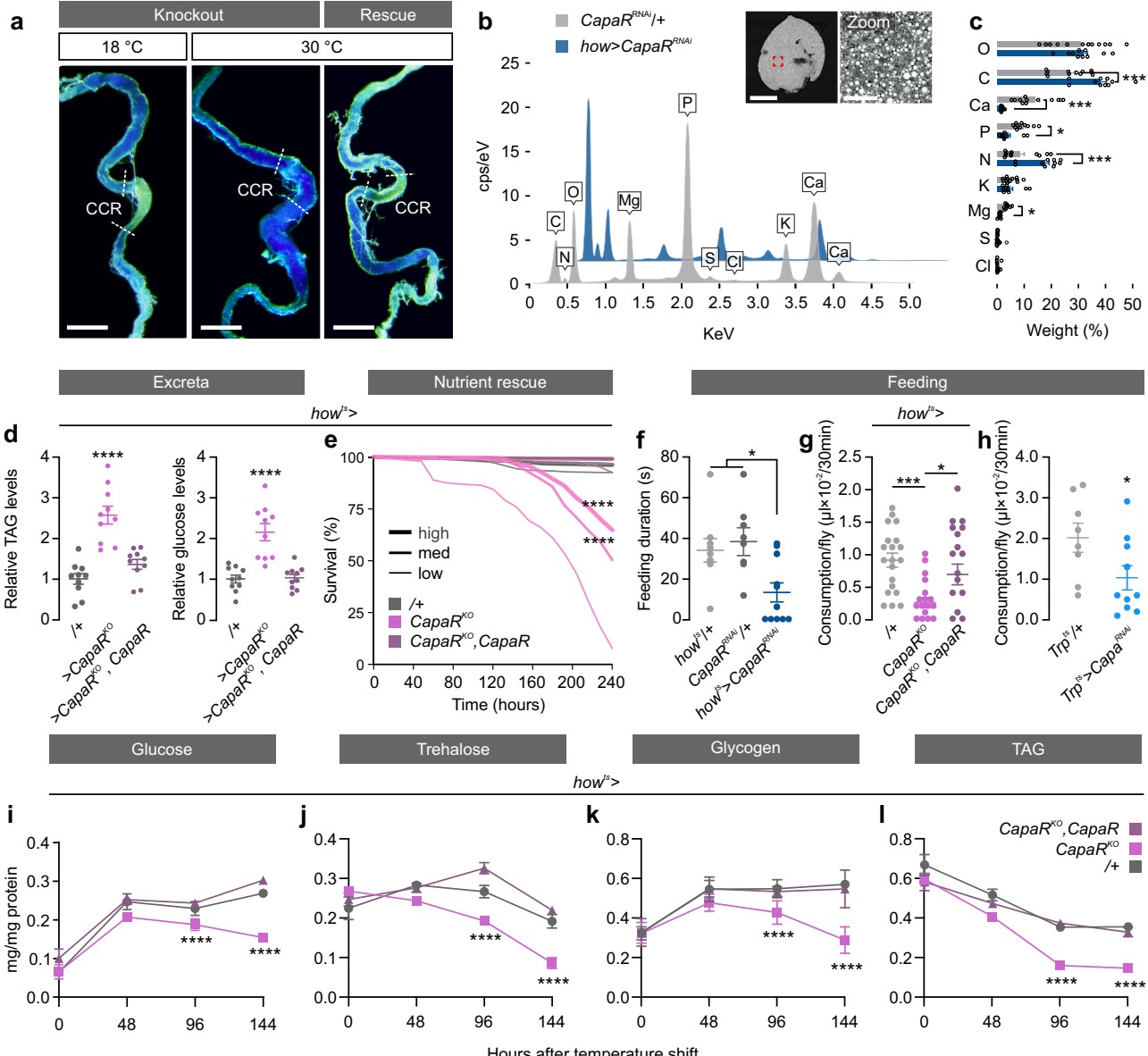

**Fig. 5 CapaR deletion in visceral muscles causes systemic metabolic imbalance. a** Dissected intestinal tracts from flies fed with food supplemented with the pH indicator BPB. A complete loss of acidity in the CCR of the midgut was observed, as evident from the absence of yellowish color, exclusively in *how^ts^ > CapaR^KO^* flies kept at the restrictive temperature (30 °C). This was not detected in animals kept at the permissive temperature (18 °C) or in rescue flies kept at 30 °C additionally carrying a *UAS-CapaR* transgene (*how^ts^ > CapaR^KO^,CapaR*). This experiment was repeated several times with similar results. Scale bar = 200 μm. **b** Elemental composition and **c** weight percentage of each element in the excreta from muscle-specific *CapaR*-silenced flies (*how > CapaR^RNAi^*; *n* = 12) relative to control (*CapaR^RNAi^/+*; *n* = 10) using scanning electron microscopy coupled with wave-dispersive X-ray analyses (mean ± SEM; two-tailed unpaired Student's *t*-test; C, *P* = 0.0007; Ca, *P* < 0.001; P, *P* = 0.002; N, *P* = 0.0018; Mg, *P* < 0.001). **d** Relative amounts of undigested triacylglyceride (TAG) and glucose levels in excreta from control (*how^ts^ > +*), knockout (*how^ts^ > CapaR^KO^*) and rescue (*how^ts^ > CapaR^KO^,CapaR*) flies raised on standard medium (mean ± SEM; one-way-ANOVA; *P* < 0.0001; *n* = 10). **e** Flies kept on media with low, medium and high nutrient concentrations show that higher nutrient concentrations reduce mortality in *how^ts^ > CapaR^KO^* (low, *n* = 325; med, *n* = 323; high, *n* = 325; *P* < 0.0001) flies. Both control (low, *n* = 272; med, *n* = 271; high, *n* = 273) and rescue (low, *n* = 251; med, *n* = 250; high, *n* = 253) flies are unaffected by the different nutrient dense diets (two-sided log-rank test; *P* < 0.0001). **f** Feeding duration in *CapaR* knockdown animals (*how^ts^ > CapaR^RNAi^*; *n* = 11) compared to parental controls (*how^ts^/+*; *n* = 9; *P* = 0.0331; *CapaR^RNAi^/+*; *n* = 8; *P* = 0.0120; mean ± SEM; one-way ANOVA). **g** Food intake in *CapaR* deficient (*how^ts^ > CapaR^KO^*; *n* = 20) flies relative to control (*how^ts^/+*; *n* = 20; *P* = 0.0008), and rescue animals (*how^ts^ > CapaR^KO^, CapaR*; *n* = 20; *P* = 0.0333) as well as in **h**. *Capa* silenced (*Trp^ts^ > Capa ^RNAi^*; *n* = 10) flies compared to control (*Trp^ts^/+*; *n* = 8; mean ± SEM; two-tailed unpaired Student's *t*-test; *P* = 0.0480). **i–l** Quantification of **i** glucose, **j** trehalose, **k** glycogen and **l** TAG levels following 48, 96 and 144 h of adult-specific transgene activation (mean ± SEM; one-way ANOVA; *n* = 10; *P* < 0.0001).

activity peaks relative to parental controls (Fig. 6a, b). This observation contrasts with the stereotypic increase in locomotion exhibited by energy-deprived wild-type animals, a behavior linked to food foraging[38]. Conversely, flies with rescued Capa/ CapaR signaling in muscles exhibited a pronounced hyperactivity

compared to control, perhaps owing to increased Capa/CapaR signaling flux in the musculature of these animals (Fig. 6a, b). To further evaluate the functions of Capa/CapaR signaling in muscles, we video-tracked individual flies with either knockdown of *CapaR* in muscles or *Capa* in neurons and quantified the total

distance traveled as well as response velocity of these animals. Our data confirmed that both muscle-specific elimination of *CapaR* as well as *Capa* depletion in Capa+ neurons cause significant reductions in total walking distance and maximal response velocity compared to controls (Fig. 6c, d). These data suggest that CapaR function in muscles is necessary for general motor activity, and that silencing Capa/CapaR signaling in muscles impairs short-term and long-term locomotor activity.

We next explored the underlying cause of this reduced activity. Capa peptides have been shown to possess myomodulatory effects in a range of insects[40–42], raising the possibility that Capa/CapaR signaling may directly regulate skeletal muscle contractility in *Drosophila* also. We therefore eliminated Capa/CapaR signaling exclusively in adult skeletal muscles (flight, leg and abdominal muscles) by using the *Act88F-GAL4* driver (Supplementary Fig. 6a–d)[43,44], to isolate the physiological effects produced by skeletal muscles. These data showed that eliminating CapaR function in the skeletal muscles failed to: (i) reduce locomotor activity, (ii) disrupt systemic energy balance apart from reducing glycogen levels; (iii) impair food intake; or (iv) induce significant mortality compared to the controls (Supplementary Fig. 6e–l). Consequently, impairing Capa/CapaR signaling exclusively in skeletal muscles does not recapitulate the full extent of phenotypes of global muscle knockout, suggesting that direct Capa-induced changes in skeletal muscle performance is unlikely to be the major cause underlying the acute fly mortality observed.

The skeletal musculature imposes large energy demands on the organism due to its large mass and high metabolic rate. We therefore asked whether reduced systemic energy levels cause the observed decline in locomotor activity. First, we assessed the tissue-specific distribution of glycogen (the main energy storage form in muscles) as a functional readout of skeletal muscle energy status using the periodic acid-Schiff (PAS) staining method[45]. Remarkably, PAS staining was almost completely abolished in the muscles and to a lesser extent in the fat body (but not in brain) of muscle-specific *CapaR* knockout flies. In addition, this effect was partially rescued by co-expressing a CapaR transgene (Fig. 6e). These data confirmed that the major glycogen storing tissues, such as the fat body and skeletal muscles, are depleted of glycogen, which together with the generally lean phenotype of muscle-specific *CapaR* knockout animals (Fig. 5i–l), imply that they are unable to meet the high energy demands of contracting muscles. In line with this hypothesis, we observed that the genes encoding the rate-limiting glycogenolytic enzymes *GlyP* and *AGL* are upregulated in *CapaR* knockout muscles, while the glycogenic enzymes *GlyS* and *AGBE* are significantly downregulated (Fig. 6f), indicating that the skeletal muscles are metabolically adapting by increasing glycogen mobilization to try and meet autonomous energy demands.

Given that muscle performance depends critically on effective calcium cycling mechanisms, and that $Ca^{2+}$-pump activity is highly sensitive to cellular energy levels[46], we rationalized that energy depletion in muscle cells may compromise $Ca^{2+}$ dynamics. To test this hypothesis, we quantified cytosolic $Ca^{2+}$ levels during steady-state and Capa-1 stimulation in muscle-specific *CapaR* knockdown animals with reduced muscle performance. Strikingly, these data showed that not only does *CapaR* depletion abolish the Capa-induced increase in intracellular $Ca^{2+}$ levels in skeletal muscles, but the unstimulated basal $[Ca^{2+}]_i$ levels were almost 50% increased relative to the control (Fig. 6g, h). This observation suggests that the transport machinery responsible for reestablishing resting $Ca^{2+}$ levels is compromised. We therefore investigated a potential role of the *Drosophila* plasma membrane calcium ATPase (*PMCA*) in muscle function—an efflux transporter crucial to maintaining resting $[Ca^{2+}]_i$ levels and for regulating the excitation-contraction coupling process in muscles[47]—and found

that targeted knockdown of *PMCA* in muscles phenocopied the dysregulated $[Ca^{2+}]_i$ levels, impaired locomotor activity and mortality phenotypes observed in muscle-specific *CapaR* knockout animals (Supplementary Fig. 6m–p). Consistent with these observations, transcript levels of *PMCA* were significantly reduced in *CapaR* knockout flies compared to control (Fig. 6i), suggesting that $Ca^{2+}$ dysregulation in skeletal muscles of muscle-specific *CapaR* knockout flies is at least partly linked with reduced PMCA activity. Overall, our data suggest that the progressive loss of muscle function observed in Capa/CapaR impaired animals is unlikely to be caused by direct Capa actions on skeletal muscles. Rather, our data point to a model in which depletion of internal energy stores reduce $Ca^{2+}$-pump activity and disrupts $Ca^{2+}$-dynamics in muscles, creating a muscular dystrophy-like phenotype analogous to that observed in human myopathies[48].

**Nutrients and water induce release of Capa peptides.** Our data imply that Capa/CapaR signaling plays a central role in coordinating postprandial homeostasis. We therefore tested if the Capa-expressing neurons are sensitive to internal signals related to water and nutrient availability. This hypothesis is consistent with the observation that Capa peptides accumulate in Va neurons during desiccation stress, but are subsequently released during recovery[49]. We thus exposed flies to either desiccation or starvation for 24 h followed by transfer to media with different water and nutrient compositions and applied complementary approaches to measure neurosecretory activity (Fig. 7a). First, we quantified *Capa* transcript levels as well as intracellular Capa prohormone levels in Capa+ neurons. Secondly, we used an in vivo calcium reporter (CaLexA) that translates sustained neural activity into GFP expression[50]. Consistent with previous findings[49], these data showed that *Capa* mRNA levels and immunoreactivity were significantly increased in Va neurons, but not SEG neurons, of desiccated $w^{1118}$ flies compared to control, yet returned to steady-state levels or lower 4 h after refeeding or rehydration (Fig. 7b, c; Supplementary Fig. 7a–c). As expected, we observed inverse changes in activity in flies expressing CaLexA-GFP in Capa-producing neurons (Fig. 7d). These data suggest that the Va neurons are less active during periods of water and/or nutrient restriction but are induced to release Capa neuropeptides at high rates following both refeeding and rehydration. Interestingly, long-term hydration of rehydrated flies led to gradual accumulation of Capa peptides, and a concomitant reduction in GFP intensity over the ensuing 24 h (Fig. 7c, d), indicating that water ingestion alone is insufficient to maintain stimulation of Capa-Va neuronal activity in hydrated flies.

Since starvation affects Capa release (Fig. 7c, d), we tested whether the Capa+ neurons were sensitive to nutritional cues. In flies exposed to nutrient-deprivation for 24 h, we observed a significant increase in *Capa* mRNA levels and Capa immunoreactivity in $w^{1118}$ flies and a parallel reduction in CaLexA-GFP activity in the Va neurosecretory cells of *Trp > CaLexA* animals, with both Capa and CaLexA-GFP signal intensities returning to steady-state levels following refeeding on standard medium (Fig. 7b–d). These data demonstrate that the Capa+ Va neurons, and again not the SEG neurons (Supplementary Fig. 7a–c), are responsive to signals related to internal nutrient availability. To further identify the dietary factors that regulate Capa release, we transferred starved flies to media that contained either sucrose or yeast as exclusive sources of carbohydrates or amino acids, respectively. Interestingly, refeeding sugar, but not yeast, reverted anti-Capa and GFP fluorescence levels similar to the ones observed in fully fed conditions (Fig. 7c, d), suggesting that the Va neurons are specifically activated in response to dietary sugars or their metabolites. To functionally test this model, we knocked down

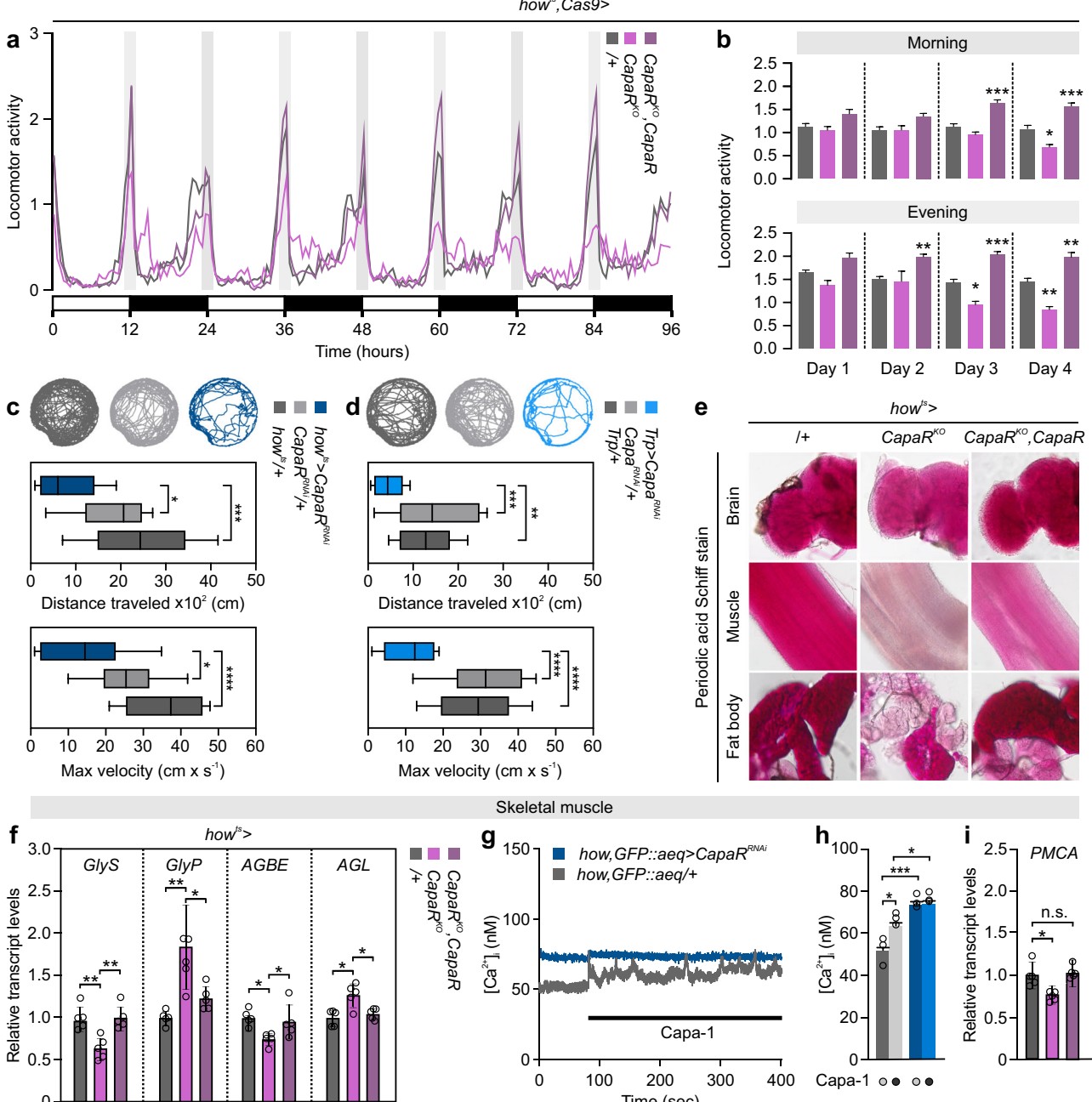

**Fig. 6 Muscle-specific *CapaR* elimination induces locomotor defects. a** Locomotor activity of individual control (/+; n = 31), muscle-specific *CapaR* knockout (*how^ts^ > CapaR^KO^*; n = 20) and rescue (*how^ts^ > CapaR^KO^, CapaR*; n = 31) flies exposed to 12-hour:12-hour, light-dark (LD) cycles for 96 h. The experiment was repeated twice with the same results. **b** Stereotypic morning and evening activity peaks measured over the experimental period (mean ± SEM; one-way ANOVA; *P < 0.05; **P < 0.01; ***P < 0.001). **c–d** Representative activity traces of video-tracked individual flies with targeted *CapaR* silencing in muscles (*how^ts^ > CapaR^RNAi^*; n = 16) or *Capa* knockdown in Capa⁺ neurons (*Trp > Capa^RNAi^*; n = 16) relative to control flies (n = 10), including quantifications of distance traveled and max velocity (Tukey's box-plots; whiskers: min and max; box: 25th and 75th percent quartiles; middle: median; one-way ANOVA; *P < 0.05; **P < 0.01; ***P < 0.001; ****P < 0.0001). **e** Periodic Acid-Schiff (PAS) staining in the brain, skeletal muscle (from thorax), and fat body showed depleted carbohydrate stores in these tissues of *how^ts^ > CapaR^KO^* flies relative to control, with partial rescue in flies additionally carrying a *CapaR* transgene (*how^ts^ > CapaR^KO^,CapaR*). **f** Transcript levels of *GlyS, GlyP, AGBE* and *AGL* relative to *RpL3* in skeletal muscles (thorax samples) from muscle-specific *CapaR* knockout (*how^ts^ > CapaR^KO^*) compared to control (*how^ts^/+*) and rescue (*how^ts^ > CapaR^KO^,CapaR*) flies (mean ± SEM; one-way ANOVA; *P < 0.05; **P < 0.01; n = 5). **g** Stereotypic cytosolic calcium traces in skeletal muscles (thorax samples) expressing the aequorin-based bioluminescent Ca²⁺-indicator (*how > GFP::aeq*, gray line), and the effect of attenuating *CapaR* expression (*how > GFP::aeq,CapaR^RNAi^*, blue line) in muscles stimulated with $10^{-7}$ M of Capa-1 peptide. **h** Bar graphs of basal and stimulated cytosolic calcium levels in skeletal muscles (thorax samples) of *how > GFP::aeq,CapaR^RNAi^* and *how > GFP::aeq* control samples (mean ± SEM; one-way ANOVA; *P < 0.05; ***P < 0.001; n = 3). **i** Transcript abundance of *PMCA* compared to *RpL32* expression in skeletal muscles from control (*how^ts^ > +*), knockout (*how^ts^ > CapaR^KO^*; P = 0.0110) and rescue (*how^ts^ > CapaR^KO^,CapaR*) flies (mean ± SEM; one-way ANOVA; n = 5).

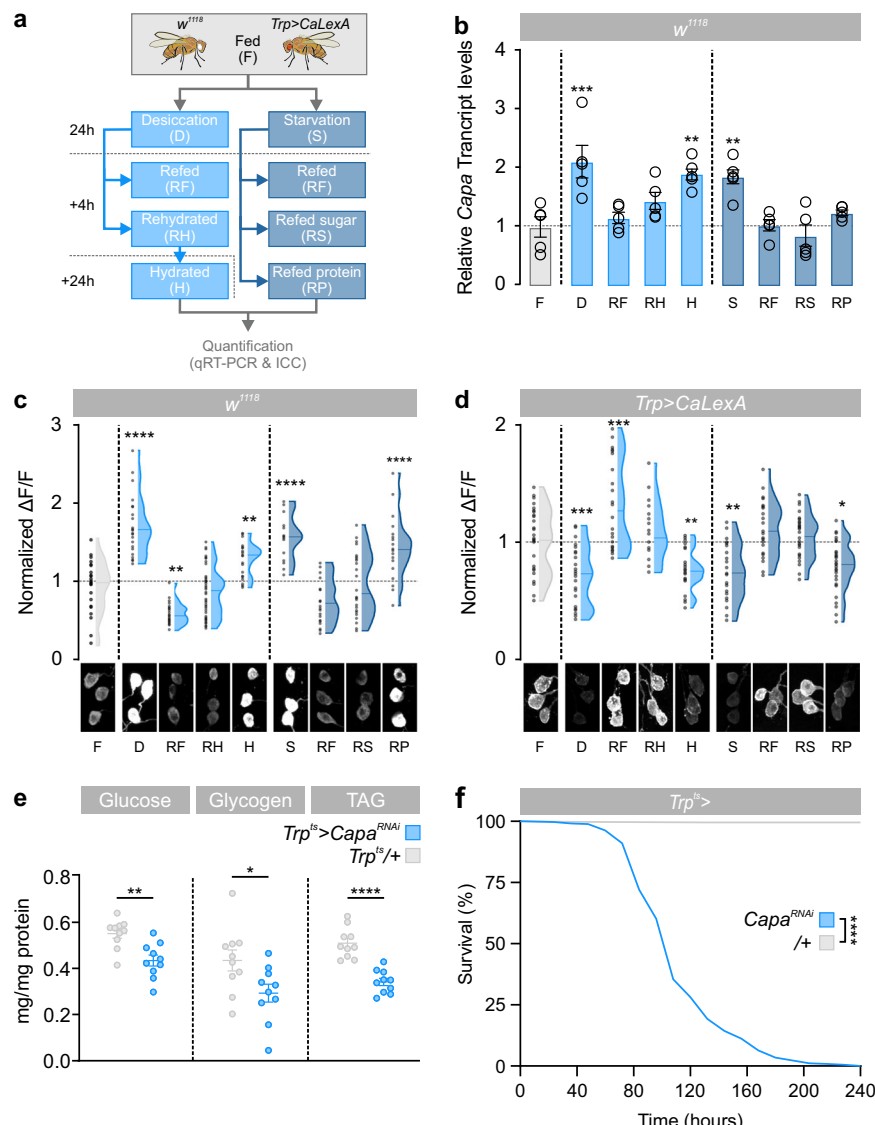

**Fig. 7 Environmental cues modulating Capa⁺ Va neuron activity. a** Experimental design for the environmental exposures of genetic background control (*w¹¹¹⁸*) and *Trp > CaLexA* flies. **b** Transcript levels of *Capa* gene expression relative to *RpL3* in Va neurons (thorax samples) of *w¹¹¹⁸* flies exposed to the environmental conditions indicated (mean ± SEM; one-way ANOVA; **$P < 0.01$; $n = 5$). **c** Violin plots and data distributions of immunofluorescence quantifications of intracellular Capa precursor levels in *w¹¹¹⁸* flies (F, $n = 23$, D, $n = 24$; RF, $n = 24$; RH, $n = 52$; H, $n = 29$; S, $n = 18$; RF, $n = 24$; RS, $n = 24$; RP, $n = 24$) and **d** CaLexA-induced GFP expression (F, $n = 30$, D, $n = 30$; RF, $n = 24$; RH, $n = 18$; H, $n = 24$; S, $n = 18$; RF, $n = 30$; RS, $n = 30$; RP, $n = 36$) in Capa⁺ Va neurons with representative images from all conditions (one-way ANOVA; **$P < 0.01$; ****$P < 0.0001$). **e** Quantification of whole animal glucose ($P = 0.0014$), glycogen ($P = 0.0292$) and TAG ($P < 0.0001$) levels following adult-specific *Capa* knockdown (*Trpᵗˢ > CapaRNAi*; mean ± SEM; two-tailed Student's *t*-test; $n = 10$). **f** Knockdown of *Capa* in adult flies (*Trpᵗˢ > CapaRNAi*; $n = 347$) results in significant mortality relative to parental control (*Trpᵗˢ > +*; $n = 409$) (two-sided log-rank test; $P < 0.0001$).

*Capa* expression specifically in the adult Capa⁺ Va neurons and assessed their ability to recover organismal energy levels following refeeding as well as their long-term survival. These data show the *Capa* silenced animals possess significantly less energy stores than control animals, suggesting that Capa⁺ neuron activation is required for reestablishing energy balance after feeding and for maintaining organismal viability (Fig. 7e, f; Supplementary Fig. 7d). Together, our data show that the different populations of Capa⁺ neurons are differentially activated: only the three pairs of Capa⁺ Va neurons relay information about the internal availability of water and sugar levels, because only these neurons release Capa peptides into the circulation in response to environmental conditions known to affect the internal abundance of these nutrients.

**Capa-mediated regulation of AKH secretion controls systemic metabolic balance.** The finding that *Capa* neurons respond directly to internal sugar levels implies that Capa peptides relay nutritional information to exert metabolic control. This prompted us to explore a potential interaction between Capa/CapaR signaling and hormonal systems controlling systemic energy balance. As in mammals, the *Drosophila* insulin-like peptides (DILPs) and AKH (a glucagon analog) act as the key counter-regulatory hormones in controlling carbohydrate metabolism[7,38,51]. Although we were unable to detect *CapaR > mCD8::GFP* expression in the adult insulin-producing cells (Fig. 2a), we surprisingly found reporter expression in the APCs in the CC; the sole site of AKH synthesis and release[38]. Furthermore, we observed specific and displaceable Capa-1-F binding overlapping with APC-targeted GFP expression

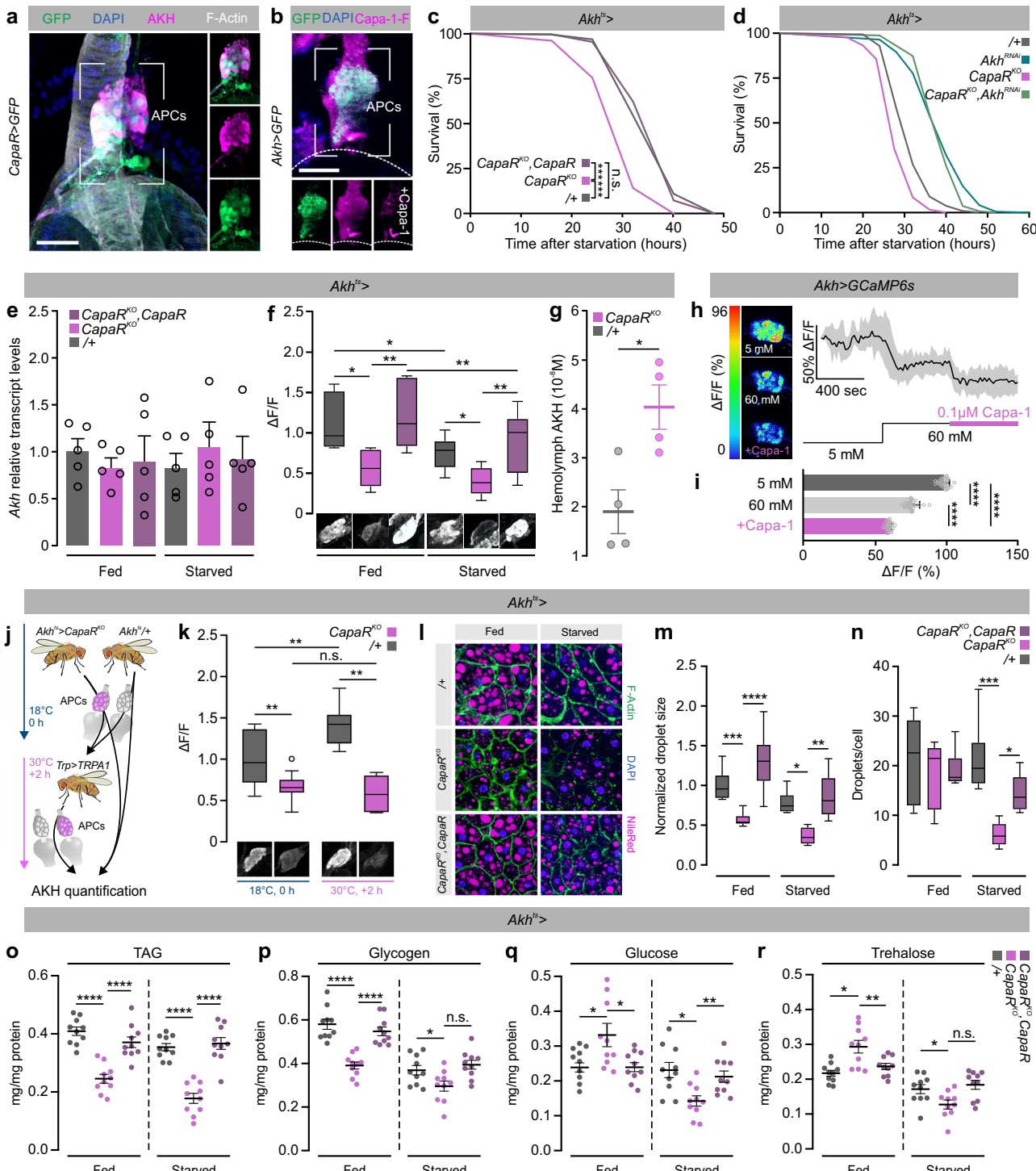

(Fig. 8a, b). These data raise the possibility that Capa peptides may regulate energy metabolism by directly modulating the AKH pathway. Because the primary function of AKH is to induce lipid catabolism, which affects organismal resistance to starvation, we measured survival under starvation in animals with reduced *CapaR* expression in the APCs to test whether Capa regulates AKH secretion. Flies with adult-specific knockout of *CapaR* in the APCs showed a marked hypersensitivity to nutrient deprivation compared to age-matched controls, a phenotype that is consistent with increased AKH release;[38] this effect was reverted in flies additionally carrying a *CapaR* transgene (Fig. 8c). In addition, we could fully

rescue the observed starvation sensitivity in *Akh^{ts} > CapaR^{KO}* flies by co-expressing an *Akh-RNAi* construct, hereby pointing to a model in which Capa/CapaR signaling directly modulates AKH secretion (Fig. 8d). Together, these data show that Capa mediates adult-specific regulation of AKH signaling, which is necessary to sustain organismal survival during nutrient deprivation.

To further probe the interactions between AKH and Capa/CapaR signaling, we compared *Akh* transcript and intracellular AKH protein levels upon APC-specific *CapaR* knockout in flies exposed to either *ad libitum* feeding or caloric deprivation. Neither starvation nor *CapaR* manipulation altered *Akh*

**Fig. 8 Systemic Capa/CapaR signaling regulates metabolic homeostasis via AKH activity. a** AKH immunofluorescence (magenta) colocalizes with *CapaR* > driven *mCD8::GFP* expression (green) confirming CapaR localization to the APCs. Scale bar 10 μm. The experiment was repeated multiple times with the same results. **b** Application of Capa1-F (magenta) shows specific and displaceable binding to *Akh > GFP* (green) positive cells in the APCs. The experiment was repeated twice with the same results. Scale bar 10 μm. **c** *CapaR* knockout in the APCs (*Akh^{ts} > CapaR^{KO}*; n = 208,) results in significantly reduced resistance to starvation compared to parental controls (*Akh^{ts} > +*; n = 213), while co-overexpression of *CapaR* (*Akh^{ts} > CapaR^{KO};CapaR*; n = 247) rescued the survival phenotype (two-sided log-rank test; (P < 0.001). **d** Knockdown of *Akh* and knockout of *CapaR* in the APCs (*Akh^{ts} > CapaR^{KO},Akh^{RNAi}*; n = 257) completely abolishes the hypersensitivity to starvation shown by APC-specific *CapaR* knockout animals (*Akh^{ts} > CapaR^{KO}*; n = 251). These animals even become starvation tolerant relative to control (*Akh^{ts} > +*; n = 314) approaching levels similar to that of *Akh* knockdown animals (*Akh^{ts} > Akh^{RNAi}*; n = 370; two-sided log-rank test). **e** Transcript levels of *Akh* relative to *RpL32* in the APCs from the different genetic backgrounds (mean ± SEM; one-way ANOVA; n = 5). **f** Immunofluorescence measurements of intracellular AKH levels in the APCs from animals exposed to either *ad libitum* feeding (fed) or caloric deprivation (starved) for 24 h, probed with an anti-AKH antibody (Tukey's box-plots; whiskers: min and max; box: 25th and 75th percent quartiles; middle: median; one-way ANOVA; Fed: *Akhts/+*, n = 6; *Akh^{ts} > CapaR^{KO}*, n = 6; *Akh^{ts} > CapaR^{KO}, CapaR*, n = 6. Starved: *Akhts/+*, n = 10; *Akh^{ts} > CapaR^{KO}*, n = 12; *Akh^{ts} > CapaR^{KO}, CapaR*, n = 15; *P < 0.05; **P < 0.01). **g** Quantification of AKH hemolymph levels from control (*Akh/+*) and *CapaR* knockout animals (*Akh^{ts} > CapaR^{KO}*) as realized by dot-blots (two-tailed Student's t-test; P = 0.0138; n = 4). **h** ΔF/F heat maps and ΔF/F traces (mean ± SD) of APCs from flies expressing the GCaMP6s calcium sensor (*Akh > GCaMP6s*) sequentially responding to low sugar (5 mM trehalose), high sugar (60 mM trehalose) and high sugar + Capa-1 (60 mM trehalose + $10^{-7}$ M Capa-1). **i** ΔF/F of APCs responding to sugar increases and Capa-1 perfusion (mean ± SEM; one-way ANOVA; P < 0.0001; n = 8). **j** Experimental design for assessing the hormonal impact of Capa signaling on AKH release. Both donor and host flies were kept at 18 °C until dissection. Once the APC-containing tissue was dissected out, the tissue were immediately transplanted into the thorax of the host flies and then kept at 30 °C for two hours to induce Capa secretion from the Va neurons. **k** Immunofluorescence measurements of intracellular AKH levels in the APCs from control (*Akh/+*; n = 11) and *CapaR* knockout animals (*Akh > CapaR^{KO}*; n = 11) before (18 °C, 0 h) and after (30 °C, +2 h) recovery of transplanted CC from *Trp > TRPA1* flies (*Akh/+*; n = 9; *Akh > CapaR^{KO}*; n = 7), as probed with an anti-AKH antibody (Tukey's box-plots; whiskers: min and max; box: 25th and 75th percent quartiles; middle: median; one-way ANOVA; n.s. P > 0.05; **P < 0.01). **l** Lipophilic dye Nile red staining of adult dissected fat bodies from fed and starved control (*Akh^{ts} > +*), knockout (*Akh^{ts} > CapaR^{KO}*) and rescue flies (*Akh^{ts} > CapaR^{KO},CapaR*). **m** Quantification of lipid droplet size and **n** lipid droplets/cell in fed and starved flies with the different genetic backgrounds (Tukey's box-plots; whiskers: min and max; box: 25th and 75th percent quartiles; middle: median; one-way ANOVA; Fed: *Akh^{ts}/+*, n = 8; *Akh^{ts} > CapaR^{KO}*, n = 6; *Akh^{ts} > CapaR^{KO}, CapaR*, n = 7. Starved: *Akh^{ts}/+*, n = 6; *Akh^{ts} > CapaR^{KO}*, n = 8; *Akh^{ts} > CapaR^{KO}, CapaR*; *P < 0.05; **P < 0.01; ***P < 0.001; ****P < 0.0001; n = 6). **o–r** Quantification of whole-body TAG, glycogen, glucose and trehalose in fed and starved animals of the different genotypes (mean ± SEM; one-way ANOVA; *P < 0.05; **P < 0.01; ***P < 0.001; ****P < 0.0001; n = 10).

expression. However, starvation induced a marked decrease in intracellular AKH levels in the APCs in all genotypes, consistent with the known induction of AKH release by starvation[38]. Moreover, in both fed and starved conditions we also observed that *CapaR* knockout in the APCs consistently lowered intracellular AKH protein levels relative to control, which was reverted to baseline levels in rescue animals (Fig. 8e, f), implying that Capa/CapaR acts to restrict AKH release into circulation. To formally assess this hypothesis, we performed a dot-blot analysis of AKH levels in hemolymph from control and APC-specific *CapaR* knockout animals. We observed a significant increase in circulating AKH levels in *Akh^{ts} > CapaR^{KO}* flies compared to parental control hereby directly coupling Capa activity to changes in the AKH hemolymph signature (Fig. 8g; Supplementary Fig. 8a). To test whether these changes in APC activity are directly mediated by Capa/CapaR signaling, we monitored calcium levels by GCaMP6s fluorescence upon Capa perfusion in ex vivo CC preparations. In line with previous reports, APC excitability decreased in response to increased trehalose levels[52,53], yet more strikingly, perfusion with Capa-1 in the micromolar range induced an additional robust GCaMP6s fluorescence decrease that was not observed upon mock perfusion (Fig. 8h, i; Supplementary Fig. 8b, c). These effects are likely mediated by systemic Capa signals in vivo, as transplantation of dissected CC (and therefore devoid of neuronal innervation) from *Akh > +* donor animals into host flies with overactivated Capa+ neurons, revealed an increase in AKH retention relative to untransplanted controls upon CC retrieval. In contrast, increased retention of AKH was not observed in CC from *Akh^{ts} > CapaR^{KO}* flies as the intracellular AKH levels in the APCs remained significantly lower compared to controls both before and after transplantation (Fig. 8j, k). Together, these data demonstrate that CapaR activation in the APCs act to repress the release of AKH into circulation, and that the Capa neuroendocrine pathway exerts metabolic control by

modulating AKH activity. Such a function is furthermore consistent with the sugar-induced release of Capa from the Va neurons given that AKH release should be repressed during sugar-replete states in order to prevent harmful hyperglycemia. However, since our data suggest Capa+ Va neuron activity is reduced, but not completely silent, during nutrient deprivation, Capa/CapaR signaling may additionally play a role in preventing AKH hypersecretion during nutrient deprived states to avoid accelerated use of internal energy stores.

Given that the main downstream action of AKH is to regulate lipolysis in the fat body, we investigated the metabolic status of *CapaR*-deficient animals by assessing their lipid reserves in the adult fat body. Using the lipophilic dye Nile Red to direct visualize lipid contents[54], we observed a significant but rescuable reduction in fat body lipid-droplet size in APC-specific *CapaR*-knockout animals under both fed and starved conditions; starvation additionally caused a decrease in lipid-droplet number, which could be partially rescued (Fig. 8l–n). Consistent with these observations, we detected a significant reduction in organismal energy reserves in both fed and starved animals compared to control, as indicated by decreased whole-body TAG and glycogen levels in APC-specific *CapaR* knockout animals (Fig. 8o, p). In addition, fed flies lacking *CapaR* in the APCs displayed a pronounced hyperglycemia relative to control and rescue animals, consistent with hyperactivation of AKH signaling in these animals. In contrast, starved flies exhibited a decrease in glycemic levels that is likely explained by the combined effects of nutrient deprivation and accelerated mobilization of internal energy stores (Fig. 8q, r), which underlies the observed hypersensitivity to starvation in *Akh^{ts} > CapaR^{KO}* animals (Fig. 8c). Altogether, our results identify a hormonal relay involving Capa+ Va neurons, which sense internal sugar availability and signal to neurosecretory cells in the CC to control AKH-mediated catabolic processes in adipose tissue to stabilize circulating energy levels in the adult fly.

## Discussion

Our work has identified a system in which Capa peptides released by six Va neuroendocrine cells of the ventral nerve cord activate their receptor CapaR localized to visceral muscles to control intestinal motility and food transit. Activating the Capa circuit increases intestinal transit causing flies to excrete large amounts of dilute deposits, whereas inactivation of CapaR in gut muscle cells results in reduced gut motility and excretion. These physiological effects are linked with the strategic localization of CapaR expression to visceral muscles in distinct regions of the gut that contain valves and sphincters[27], implying that Capa peptides are part of a digestive program by which the transit of intestinal contents is sensed and modulated. Consistent with this idea, genetic inhibition of gut peristalsis and intestinal passage appears to increase the amount of fluid absorbed by the gut, which leads to a concentration of waste products and causes a constipation-like phenotype analogous to the effects observed in humans suffering from gastrointestinal motility disorders[30]. This manipulation also frequently associates with intestinal distention, which incidentally helps explain the reduced food intake by these flies, given that enteric neurons convey stretch-dependent feeding-inhibitory signals[55,56]. Our work and previous studies[11,49] are therefore consistent with a model in which the synchronized control of renal secretion and gut motility by the Capa circuit helps synergize the homeostatic control of water balance and waste excretion, which shows striking functional homology to the actions of NmU signaling in humans[13]. Intriguingly, the coupled control of renal and gut functions appears to have been co-opted during evolution since other hormones exerting systemic control of diuresis have also been reported to modulate feeding behavior and intestinal contractions[6,16,56,57]. The convergence of multiple neuropeptide signaling pathways on organ functions critical to controlling postingestive physiology suggests a fundamental need for coupling food intake with fluid and waste excretion. How these endocrine networks interact to regulate these vital processes remains to be examined.

In addition to compromising intestinal fluid balance and excretion, we also found that silencing Capa-mediated visceral contractions disrupt the integrity of the midgut acidic zone. The acidic zone consists of acid-secreting copper cells that are functionally related to the gastric parietal cells of the mammalian stomach, and this region is necessary to facilitate the digestion and absorption of macronutrients and metals as well as for sustaining adult survival[31,34,58,59]. Although we cannot exclude that the loss of acidity is due to impaired copper-cell acid secretion, our data suggests that CapaR-expressing visceral muscles at the anterior and posterior junctions of this region serve a sphincter-like role that is necessary to maintain gut compartmentalization and optimal pH of the CCR; a role strikingly similar to that played by the esophageal and pyloric sphincters in humans. Such a function would further contribute to shielding neighboring gut regions from the low pH of the CCR. This idea is consistent with the morphological constrictions of the gut tube at either end of the acidic zone and is supported by a recent study demonstrating that ablation of enteroendocrine-derived DH31, which controls food passage into the acidic zone in *Drosophila* larvae, leads to inappropriate mixing of acidified and non-acidified gut contents[58,60]. Our data thus point to a mechanism in which nutrient-dependent Capa release helps maintain functional gut compartmentalization in order to promote and optimize nutrient digestion and uptake following food consumption, which is necessary to maintain adult fly survival.

Silencing Capa/CapaR signaling in muscles induces a complex metabolic phenotype characterized by lipodystrophy and hypoglycemia accompanied by hypophagia, locomotor deficits, and reduced organismal lifespan. How can we reconcile these phenotypes? Several lines of evidence indicate that impaired nutrient uptake is a conserved mechanism reducing organismal longevity[61,62], raising the possibility that these phenotypes arise as a consequence of malnutrition and poor intestinal absorption. This idea is consistent with the gradual decrease in whole-body energy stores following conditional *CapaR* knockout in visceral muscles and is further supported by the observation that survival can be enhanced by exposing these flies to higher nutrient concentrations. Concordantly, our data show that the loss of internal metabolic stores correlates with the progressive decline in physical activity. These results may imply that the age-dependent decline in locomotor activity derives from impaired energy homeostasis. Intriguingly, locomotor deficits are known to impair foraging behavior and food intake[63], suggesting that the reduced muscle performance observed in Capa/CapaR deficient animals may contribute to the hypophagia exhibited by these animals, causing further malnutrition and ultimately death. Altogether, our data support a model in which nutrient absorption is one of the key aspects of Capa-mediated control of adult metabolic homeostasis, which is essential to sustain physical activity, feeding and survival.

Our study suggests that the Capa+ Va neurons, but not the SEG neurons[10,22], secrete Capa peptides in the presence of high internal water and sugar levels, while desiccation and caloric deprivation reduce Capa hormone release. These data connect homeostatic deviations in internal water and sugar availability to Capa/CapaR signaling, and remarkably show that the Capa Va and SEG neurons are activated by different physiological signals. Yet, how might this type of regulation be adaptive? One possibility is that food and water consumption change the internal metabolic status of animals, which requires appropriate activation of specialized organs to restore homeostasis. For example, feeding in *Drosophila* is associated with fluid uptake, which needs to be balanced by renal excretion to avoid acute effects; prolonged inactivation of diuresis can cause severe fluid retention, resulting in cuticle rupture and death[27]. Conversely, nutrient and water deprivation induces osmotic perturbations requiring compensatory actions to reduce organismal water loss; inhibiting tubule and intestinal activities promotes fluid retention and increase survival under desiccating conditions[49,64,65]. Our study thus provides evidence of a mechanism by which homeostatic deviations in internal osmolality may be reported by the Capa+ Va neurons, which provide systemic feedback regulation on intestinal and renal activities to restore fluid and ion balance.

Our finding that Capa+ Va neurons are potently activated by high hemolymph sugar levels and that Capa modulates AKH secretion surprisingly implicate Capa action in glycemic control. Our results consistently show that impaired Capa/CapaR signaling in the APCs due to *CapaR* elimination induces a marked hypersensitivity to starvation; a phenotype that is also associated with a significant increase in AKH secretion, reduced fat body energy reserves, and hyperglycemia. Whereas the impact of insulin regulation on carbohydrate metabolism is well studied in both humans and flies, less is known about AKH/glucagon. Our study reveals a neuroendocrine mechanism in which Capa peptides released from the CNS exert metabolic control by providing feedback regulation on AKH-mediated lipolysis in the fat body to stabilize circulating sugar levels. Intriguingly, similar suppression of a metabolic hormone was recently reported for NmU signaling in mammals[66], implying that vertebrates and invertebrates may share evolutionarily conserved mechanisms for regulating sugar metabolism.

In sum, our work has uncovered a key role for Capa/CapaR signaling in coordinating organ-specific post-prandial responses that are essential to sustaining adult fly survival. This inter-organ signaling system consists of a CNS/gut/renal/fat body signaling

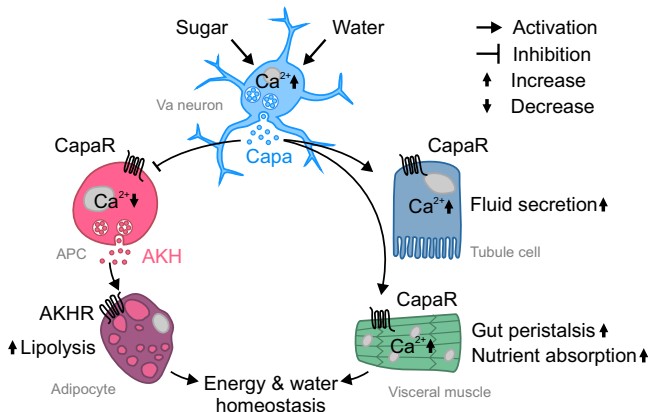

**Fig. 9 Model for the systemic actions of Capa/CapaR signaling in mediating nutritional and osmotic homeostasis in adult *Drosophila*.** Capa+ Va neurons respond to internal sugar and osmotic cues by activating two separate pathways: one stimulating renal and intestinal activities to promote fluid secretion, gut contractions and nutrient absorption, and another to restrict glucagon-like AKH release from the APCs and subsequent lipolytic activity in adipose tissue. The Capa circuit thus couple internal cues related to sugar and water abundance to homeostatically regulate energy and water balance.

module, which operates via a discrete population of sugar- and osmosensitive Capa+ neurons that release Capa peptides into circulation to control gut peristalsis, nutrient uptake, renal secretion and fat body lipolysis, to ultimately promote internal osmotic and metabolic homeostasis following sugar and water ingestion (Fig. 9). Given the remarkable functional conservation between insect Capa and mammalian NmU signaling highlighted above, and the emerging understanding of NmU as a key endocrine regulator of energy homeostasis in humans, our work may help shed new light on the pathophysiology underlying human metabolic disorders such as diabetes and obesity.

## Methods

**Fly stocks**. *UAS-CapaR^RNAi#2* (27275), *UAS-mCD8::GFP* (5137), *UAS-mLexA-VP16-NFAT* (66542), *UAS-GCaMP6s* (42749), *UAS-TRPA1* (26263), *UAS-tDH44* (80702), *Lab-GAL4* (43562), *elav-GAL4* (458), *Cg-GAL4* (7011), *201Y-GAL4* (4440), *c564-GAL4* (6982), *how(24B)-GAL4* (1767), *Akh-GAL4* (25684), *Tubulin-GAL80^ts* (7019), *Act88F-GAL4* (38461) and *Trp-GAL4* (49296) flies were obtained from Bloomington *Drosophila* Stock Center (Indiana University, Bloomington, USA). Moreover, *UAS-CapaR^RNAi#3* (105556), *Capa^RNAi* (101705) and *PMCA^RNAi* (101743) transgenic fly lines were obtained from the VDRC RNAi stocks (Vienna *Drosophila* Resource Center, Austria). *Tinman-GAL4* was kind gift from Dr. Manfred Frasch (University of Erlangen-Nuremberg, Germany) and *Hand-GAL4* was kind gift from Dr. Achim Paululat (University of Osnabrück, Germany). The *CapaR-GAL4*, *CapaR-GAL4;UAS-GFP::aequorin*, *how(24B)-GAL4;UAS-GFP::aequorin*, and *Uro-GAL4* drivers were previously generated in-house[11,17]. In addition, a viable doubly homozygous *CapaR^RNAi* (2xRNAi) line, targeting the same region of the *CapaR* gene as previously described[11], was generated using standard genetic techniques. All fly lines were maintained on a standard *Drosophila* cornmeal medium at 25 °C, 50-60% humidity under a 12-hour:12-hour light-dark photoperiod. Standard fly crosses were performed at 25 °C. Flies carrying the *TubulinGAL80^ts* transgene encoding a temperature-sensitive transcriptional repressor were raised at 18 °C and subsequently sorted for males and females upon eclosion. After sorting, flies were kept for another 24 h at 18 °C before being switched to 30 °C for transgene induction and maturation for 4 days unless otherwise specified. Only male flies were used for experimentation. All fly lines used and made in this study were backcrossed to *w^1118* to minimize genetic variations. Therefore, *w^1118* was used as genetic background control.

**Drosophila transgenesis**. The ORF of *CapaR* (*CG14575*) was amplified from whole *w^1118* fly cDNA as template and cloned into pUAST attB vector using *Not*I and *Kpn*I whose recognition sites are included on the primers (Supplementary Table 1), then integrated on the second chromosome by site-directed insertion using the plasmid encoding phiC31 integrase and an attP landing site carrying recipient line, *y¹ w^1118; PBac{y+-attP-9A VK00018}*[67] (Bloomington *Drosophila* Stock Center #9736). The tissue-specific *CapaR* CRISPR^KO construct was cloned

into pCFD6 vector (www.crisprflydesign.org, Addgene #73915) according to the website's instruction. In this vector, we cloned four independent gRNA constructs designed to target non-coding regions (introns and 3′UTR; see primers listed in Supplementary Table 1) of the *CapaR* genomic DNA sequence. After sequencing, the construct was inserted on the third chromosome in a recipient line carrying phiC31 integrase and an attP2 landing site; *y,w, P(y+.nos-int. NLS); P(CaryP)attP2* (gift from Dr. Diogo Manoel, Instituto Gulbenkian de Ciência, Portugal). Germline transformation was carried out in-house.

**Survival assay**. Flies from each genetic background were collected within 24 h after eclosion and maintained on standard *Drosophila* cornmeal medium (app. 30 flies per vial) at either 18, 25 or 30 °C at 50–60% humidity with a 12-hour:12-hour light-dark photoperiod. In all conditions, survivors were transferred to new vials every three days. Each vial was then observed for dead flies (no movement after gently tapping the vial) every 24 h for 10 days. Data was expressed as per cent survival over time with 213–793 flies counted for each genotype and condition.

**Quantitative real-time (RT) PCR analysis**. Total RNA was extracted from whole animals or dissected tissues using the RNeasy Mini Kit (QIAGEN) with DNase treatment to remove contaminating genomic DNA. RNA was quantified using a NanoDrop spectrophotometer (Thermo Scientific) and reverse transcribed into cDNA using iScript Reverse Transcription Supermix (Bio-Rad). Quantitative real-time PCR (qPCR) was performed using the QuantiTect SYBR Green PCR Kit (Qiagen) and the Mx3005P qPCR System (Agilent Technologies) with transcript levels normalized to *RpL32* or *RpL3* reference genes and expressed as fold change compared to controls ± SEM (*n* = 5). The primers used are listed in Supplementary Table 2.

**Whole fly paraffin-embedded sections and immunostaining**. Paraffin sections were made according to a modified protocol previously described[68]. In brief, *CapaR > GFP* flies were fixed in 4% (wt/vol) paraformaldehyde (PFA) for 30 min, and then dehydrated through a graded series of ethanol and xylene, before being embedded in paraffin. Next, ~8 μm thick paraffin sections were cut on a Leica RM 2235 microtome (Leica Biosystems, Nussloch, Germany), and the sections were transferred to individual glass microscope slides for subsequent use. For immunostainings, the sections were deparaffinized in Histo-Clear (National Diagnostics, USA) for 15 min, rehydrated through a graded series of alcohol (99%, 90% and 70% for 10 min each step), and rinsed in PBS. The sections were then incubated in blocking buffer (PBS with 0.1% Triton X-100 and 2% normal goat serum), containing polyclonal Alexa Fluor 488-conjugated rabbit anti-GFP (1:200; Thermo-Fisher, #A21311) and DAPI (4′,6-diamidino-2-phenylindole, 1 μg ml⁻¹ ThermoFisher, #D1306) overnight at 4 °C. Finally, the sections were washed repeatedly in PBS and mounted in Vectashield (Vector Laboratories Inc, CA, USA). Image acquisition was performed on an inverted Zeiss LSM 880 confocal laser-scanning microscope (Zeiss, Oberkochen, Germany) and processed with CorelDraw X8.

**Immunocytochemistry**. Immunocytochemistry was performed as previously described[69]. In brief, *Drosophila* tissues were dissected in Schneider's insect medium (ThermoFischer Scientific, UK), fixed in 4% (wt/vol) paraformaldehyde in PBS for 30 min at room temperature and washed twice for 1 h in blocking buffer. Tissues were then incubated in primary antibodies in blocking buffer overnight, followed by overnight incubation in secondary antibodies in blocking buffer at 4 °C. Counter staining with DAPI and/or Rhodamine-coupled phalloidin (1:100; ThermoFisher, #R415) was performed where appropriate. The primary antibodies used were DyLight 488-conjugated goat anti-GFP antibody (1:500, ThermoFisher, 600-141-215), Alexa Fluor 488-conjugated mouse anti-GFP (1:500, Invitrogen, A11120), rabbit anti-CapaR[11] (1:500), rabbit anti-Capa precursor peptide[9] (1:500), rabbit anti-AstC, rabbit anti-CCHa1, rabbit anti-DH31, rabbit anti-NPF, rabbit anti-TK[70,71] (1:100; generous gifts from Prof. Jan Veenstra, University of Bordeaux, France), mouse anti-Prospero (1:50; Developmental Studies Hybridoma bank) and rabbit anti-AKH (1:500; raised against the secreted portion of the peptide[38]), a generous gift from Dr. Jae Park, University of Tennessee, US. The primary antibodies were visualized with goat anti-rabbit Alexa Fluor 488, 555 or 594 (1:500; ThermoFischer #A32731, #A21429 or #R37117). For quantifications of intracellular Capa and AKH peptide levels, image acquisition was performed using identical microscope settings within each experimental setup as previously described[72]. For staining of intracellular lipid droplets in the fat body, adult adipose tissue was fixed as described above and the lipophilic dye Nile red (2.5μg/ml) was applied for 30 min after which tissues were rinsed with PBS. All samples were mounted on poly-L-lysine coated 35 mm glass bottom dishes (MatTek Corporation, MA, USA) in Vectashield. Retained intracellular Capa prohormone and AKH peptide levels as well as lipid droplet size and number were calculated using the Fiji (2.1.0) software package.

**CC transplantation experiments**. CC-containing tissues (a small piece of esophagus including the proventriculus) from both control (*Akh^ts > +*) and *CapaR* knockout (*Akh^ts > CapaR^KO*) donor animals were dissected, with one half of the samples fixed immediately, and the other half transplanted into animals expressing

the heat-sensitive $K^+$ channel, TRPA1, in Capa+ neurons (*Trp > TRPA1*), in order to generate heat-induced Capa titers under *in vivo*-like conditions. Briefly, the host fly was knocked out using $CO_2$, the wings were removed, and a small incision was made along the dorsal border between the scutum and scutenum of the thorax using a small piece of a double edged carbon steel laser blade (cat#200430; Electron Microscopy Sciences, PA, US), after which the CC-containing tissue was transplanted into the incision using an in-house modified fine forceps (Dumont, FL, US). This procedure if performed correctly is associated with negligible mortality. Following 2 h incubation in the transplanted flies at 30 °C, the transplanted CC were recovered and probed for anti-AKH activity as described above.

**Dot-blot assay.** Hemolymph was harvested from 200 flies and pooled into one sample, with four biological replicates collected per genotype, according to a modified protocol previously described[65]. From each sample, 5 μl of hemolymph was dropped on a 0.2 μm nitrocellulose membrane (GE Healthcare) and left to dry at RT for 20 min. In addition, successive dilutions of known AKH concentrations (AKH peptide was a generous gift from Dr. Dalibor Kodrík, Biology Centre CAS, Czech Republic) were dropped on a separate membrane to allow estimation of sample AKH levels. Both membranes were incubated in the same petri dish at all times hereafter. Next, the membranes were fixed in 4% PFA in PBS for 20 min and then blocked with 5% bovine serum albumin (BSA) in PBS containing 0.1 Tween-20 (PBST/BSA) for 1 h at RT, and finally incubated in rabbit-anti AKH (1:1000) in PBST/BSA overnight at 4 °C. Following several washes in PBST, the membranes were then incubated in a goat-anti-rabbit IR800 secondary antibody (1:2500; LI-COR, 926-32211) in PBST/BSA for 1 h at RT and then imaged on an Odyssey Fc imaging system (LI-COR). The integrated densities of the black dots followed by background subtractions were considered as amounts of AKH in the sample. The quantification and analysis of the dot-blot results were conducted using the Fiji (2.1.0) software package, and the AKH values were estimated from blotting known concentrations (see Supplementary Fig. 8a for source data) using GraphPad Prism 9.1.

**Peptide synthesis and ex vivo receptor-binding assay.** *Drosophila* Capa-1 and Capa-2 (both with and without an N-terminal cysteine) were synthesized by Cambridge Peptides (Birmingham, UK), and subsequently coupled to high quantum yield fluorophores via a cysteine-linker to make fluorescent TMR-$C_5$-maleimide-GANMGLYAFPRVamide (Capa-1-F), and Alexa488-$C_5$-maleimide-ASGLVAFPRVamide (Capa-2-F). Fluorescent Capa peptides were applied to tissues in an ex vivo receptor-binding assay as previously described[16]. In brief, tissues of interest were dissected from cold anesthetized animals and mounted on poly-L-lysine coated glass bottom dishes before being setup in a matched-pair protocol. One batch was incubated in 1:1 (vol/vol) mix of *Drosophila* saline (NaCl 117.5 mM, KCl 20 mM, $MgCl_2$ 2 mM, $CaCl_2$ 2 mM, $NaHCO_3$ 10.2 mM, $NaH_2PO_4$ 4.5 mM, HEPES 8.6 mM, Glucose 20 mM, pH 6.8) and Schneider's medium (artificial hemolymph; AHL) containing DAPI, while to the other batch was additionally added $10^{-7}$ M of Capa-1-F. The batch without the labeled peptide was used to adjust baseline filter and exposure settings to minimize background during image acquisition on a Zeiss LSM 880 confocal microscope. Specificity of binding was additionally verified by competitive displacement of the labeled ligand with unlabeled peptide at a concentration of $10^{-5}$ M.

**Real-time measurements of intracellular calcium changes.** Cytosolic calcium measurements were performed according that previously described[21]. Tissues were dissected from flies expressing *UAS-GFP::aequorin* driven by *CapaR-GAL4* or *how-GAL4*. For each sample, 20–30 live intact adult tubules, proventriculus, midgut, hindgut, brain, gonads, salivary gland, legs or skeletal muscles (thoraces) were transferred to 5 ml Röhren tubes (Sarstedt AG & Co., Nümbrecht, Germany) in 175 μl Schneider's medium and subsequently incubated in the dark with 2 μl coelenterazine in ethanol (final concentration of 2.5 μM) for 3 h to reconstitute active aequorin. Real-time luminescence was measured on a Berthold Lumat LB 9507 luminometer (Berthold technologies, Bad Wildbad, Germany). A stable baseline was established prior to both mock injection and subsequent injection with Capa-1 peptide (final concentration $10^{-7}$ M), and the luminescence was measured in the ensuing period. After each experiment, undischarged aequorin was measured by permeabilizing the cells with 300 μl lysis buffer (1% Triton X-100 and 0.1 M $CaCl_2$). Real-time calcium concentrations ($[Ca^{2+}]_i$) throughout the experiments were then back-calculated with an in-house PERL routine.

**Gut motility assay.** Guts from control and *CapaR* knockdown flies were dissected in artificial hemolymph (AHL) with minimal disruption of attached tissues and without removing the head. Individual exposed guts were next pinned onto a Sylgard-lined petri dish, with fine tungsten pins through the proboscis and a small piece of cuticle attached to the end of each gut and bathed in 100 μl of AHL. Following a 5-minutes acclimation period, each gut was then recorded with a Leica IC80 HD camera mounted on a Leica M50 stereomicroscope for 3-minutes, before the solution was exchanged with 100 μl of AHL and recorded for another 3-minutes. This was done to correct for potential artefacts on gut contraction frequency associated with the exchange of solutions. Next, the AHL solution was replaced with 100 μl of $10^{-7}$ M of either Capa-1 or Capa-2 peptide in AHL, and

tissues recorded for an additional 2 × 3-min. Contractions of the entire gut were visually counted post image acquisition, and the number of contractions was normalized to the number of contractions observed in the initial AHL solution with $n = 6–8$ guts counted for each genotype.

**Calcium imaging in the APCs.** For GCaMP6s imaging of sugar and Capa responses, foreguts including intact CC were dissected in HL3.1 AHL[73] and immobilized by mounting in poly-L-lysine coated dishes. Following an acclimation period of 30 min, CC were imaged on a Zeiss LSM 900 using a 488 nm laser line with a 20x air objective (NA = 1) and an image rate of one z-stack of 10–15 sections (512 × 512 pixels) every 20 seconds. The serial application protocol consisted of 0–10 min in low sugar AHL (5 mM trehalose, 115 mM sucrose), 10–20 min in high sugar AHL (60 mM trehalose + 60 mM sucrose) and 20–30 min with $10^{-7}$ M Capa-1 in high sugar AHL. Quantifications were performed on maximum projection z-stacks for each time point using the Fiji (2.1.0) software package and the data expressed as %ΔF/F = $((F_t/F_0)/F_0)$ where $F_0$ is the fluorescence of first time point and $F_t$ is the fluorescence of each subsequent time point.

**Quantitative analysis of defecation behavior.** Analysis of defecation behavior was performed according to a modified protocol[27]. Standard *Drosophila* food was prepared and allowed to cool to roughly 65 °C before it was supplemented with 0.5% (wt/vol) Bromophenol blue (BPB) sodium salt (B5525, Sigma). Both gut transit and excreta quantification experiments were performed on batches of 10 flies, which were placed in standard fly vials or 50-mm Petri dishes containing BPB-dyed food. For gut transit measurements, the flies were transferred to fresh vials every hour and the number of excreta manually scored. For Petri dishes, digital images of Petri lids were obtained using a photo scanner (Konica C454e), which were processed in Fiji (2.1.0) before being analyzed (including deposit number, size, and dye intensity) using the T.U.R.D – the ultimate reader of dung (version 0.8) software package[26]. Dissections of dye-containing intestines were performed in AHL, leaving the head and posterior cuticle intact to prevent leakage of gut contents.

**Quantitative elemental microanalysis.** Deposits were collected and transferred to individual aluminum stubs. Samples were then examined in a Zeiss Sigma variable pressure analytical scanning electron microscope (Carl Zeiss, Oberkochen, Germany) in combination with AZtecEnergy microanalysis software (Oxford instruments, Oxford, UK). Element microanalysis and acquisition of composition maps was carried out using high accelerating voltages (≥20 kV) in combination with electron backscatter (BSEM) and both energy-dispersive X-ray spectroscopy (EDS) and wavelength-dispersive X-ray spectroscopy (WDS), providing both structural and quantitative measures for excretion composition at high resolution.

**Feeding assays.** Groups of flies were starved for 4 h, and then transferred to normal food containing 0.5% BPB for another 30 min, followed by freezing the flies at −80 °C. Groups of five flies were transferred to 2 mL tubes and were homogenized in 100 μl PBS + 0.1% Triton X-100 using a TissueLyser LT (Qiagen). The samples were then spun down at 12,000 rpm and the supernatant transferred to fresh vials and the absorbance of each sample at 603 nm was measured using a NanoDrop. The resulting absorbance values were then subtracted the mean background value obtained from unfed flies ($n = 10$), and the data were normalized to control. A total of $n = 22–27$ samples were measured for each genotype.

For analyzing feeding behavior in individual flies, animals were starved for 4 h, before being placed in a *Drosophila* breeding chamber separated by a divider. Following a 5 min acclimation period, the divider was removed allowing the fly access to small drop of 5% (wt/vol) glucose in in $ddH_2O$. The chambers were filmed in the ensuing 10 min period, and the time spent in contact with food during this period was quantified post acquisition with $n = 10–12$ for each genotype.

**Nutrient rescue.** An artificial diet chiefly consisting of a yeast-sugar medium was used for a nutritional rescue assay. The medium consisted of 80 g/l yeast and 90 g/l sucrose boiled with 1% agar, 4.8% propionic acid and 1.6% methyl 4-hydroxybenzoate (all components from Sigma). Media with lower nutrient concentrations were made by keeping sucrose concentration constant but reducing yeast concentrations to 16 g/l and 0 g/l, respectively. Up to 30 animals were transferred into vials containing the appropriate medium, and fly mortality was counted every 12 h for the ensuing 10 days with a total of $n = 251–323$ flies used per condition.

**Nutrient-level assays.** Four days after transfer to the restrictive temperature (30 °C), flies were collected and frozen at −80 °C in groups of ten animals, and subsequently homogenized in 300 μl PBS + 0.5% Tween-20 (PBST) using a TissueLyser LT (Qiagen). Next, 2 μl was used to quantify protein content using the Bradford assay for subsequent normalization of nutrient levels. The samples were then heat-inactivated at 70 °C for 5 min, centrifuged to pellet debris for 3 seconds at full speed, before the supernatant was transferred to fresh vials. For triacylglyceride measurements, 2 μl of the supernatant was incubated with 4 μl Triglyceride Reagent

(Sigma, T2449) and 6 µl PBST at 37 °C for 30 min; next, 20 µl Free Glycerol Reagent (Sigma, F6428) was added, and the reaction was incubated at 37 °C until color development occured (usually 30–60 min). Absorbance at 540 nm was then measured. For glycogen measurements, 1 µl homogenate was mixed with 0.2 µl (0.06 U) of low-glucose Amyloglucosidase (Sigma, A7420-5MG) in 9.8 µl of PBST and incubated at 37 °C for 30 min; Glucose Oxidase (GO) reagent (25 µl) from the Glucose (GO) Assay Kit (Sigma, GAGO20) was added to determine total glucose + glycogen. Another 2 µl homogenate was mixed with 25 µl GO reagent alone (without the Amyloglucosidase) to determine free glucose. To determine trehalose contents, 1 µl supernatant was incubated with 0.5 µl Trehalase from porcine kidney (Sigma, T8778) with 9.5 µl PBST and incubated at 37 °C overnight. Glucose Oxidase (GO) reagent (25 µl) was then added and incubated at 37 °C to determine trehalose + free glucose. Free glucose readings were subtracted from measurements of glycogen + glucose and trehalose + glucose to obtain readings for glycogen and trehalose, respectively. All absorbances were measured using an EnSight multi-mode plate reader (PerkinElmer), and values were normalized to protein content.

To measure the lipid and glucose contents of excreta, 3 days after temperature shift, flies were incubated with 0.5% BPB-supplemented fly medium for an additional day for prefeeding. Then, groups of five flies were transferred to 2-ml tubes containing BPB food cast into the lid of the tube for an additional 24 h at 30 °C. After carefully removing the animals and cutting of the lids, tubes were washed with 100 µl of PBST, gently agitated with fresh lids and read at 540 nm absorbance for normalization. To measure the lipid and glucose contents of excreta, 10 µl of extracts were used and quantified as described above.

**Locomotor assays**. To analyze long-term locomotor activity of individual flies, we used the *Drosophila* activity monitor system (TriKinetics) in combination with the pySolo software package[74] and a custom MATLAB script (MATLAB R2016b, The MathWorks Inc, Natick, Massachusetts) as previously described[75]. Individual 4-day old flies of the appropriate genotype were housed in 65-mm-long glass tubes containing food (5% sucrose and 2% agar in water) in one end and a cotton plug in the other, which were assembled into the monitoring system. The experiments were run in a behavioral incubator at 25 °C with 50–60% humidity and a 12-hour:12-hour light-dark circadian rhythm. The flies were allowed to acclimate for 14 h prior to data acquisition. The activity of individual flies was then measured as the number of beam crossings per 10 min, which was calculated using pySolo software in combination with an in-house MATLAB script. The stereotypic early activity period (EAP) and late activity period (LAP) were used to assess significant changes in locomotor activity.

We also analyzed short-term parameters of locomotor activity using a custom-built activity monitor system. Flies ($n = 10$–16) were isolated individually in Plexiglas chambers, and allowed to acclimate for 2 h. Flies were then recorded at the same time for 5 min using a high-resolution digital camera. Following an experimental cycle, parameters such as activity traces, total distance traveled and the maximal response velocity for each individual fly was calculated in MATLAB.

**Histochemistry**. Adult brains, skeletal muscles (from thoraces), and fat bodies were dissected in PBS and fixed in 4% PFA in PBS for 20 min. Next, tissues were washed twice in PBS, incubated with periodic acid solution (Merck #395B-1KT) for 5 min, and washed twice in PBS. Samples were then stained with Schiff's reagent (Merck #395B-1KT) for 15 min, washed twice in PBS, and mounted in 50% glycerol in PBS. Images were acquired with an BX-51 microscope equipped with a digital camera (Olympus). Two independent experiments were conducted and gave the same results.

**Starvation resistance**. Flies were raised at the permissive temperature (18 °C) until 24 h after eclosion, after which they were transferred to the restrictive temperature (30 °C) to disinhibit GAL4 activity. Three days after transgene activation at 30 °C, flies were transferred on to 1% non-nutritive agar and incubated at 30 °C. Each vial was then observed every 8 h until all flies were counted as dead.

**Ramsay fluid secretion assay**. Secretion assays were performed as described previously[69]. Intact Malpighian tubules from 7-day-old female flies were dissected in AHL and isolated into a 10 µl drop of a 1:1 mixture of AHL and *Drosophila* saline. The tubules were left to secrete for approximately 30 min, with non-secreting tubules being replaced if necessary, to produce a set of 10–20 working tubules. A drop of secreted fluid was subsequently collected every 10 min, and the diameter was measured using an eyepiece graticule. The volume of each droplet was calculated as $4\pi r^3/3$, where r is the radius of the droplet, and secretion rates were plotted against time. Secretion was measured under basal conditions in order to establish a steady rate of secretion, prior to stimulation with labeled or unlabeled Capa peptides, with an increase in fluid secretion rate being taken as an indication of a diuretic effect.

**Statistics**. Numerical data are presented as means ± SEM The normal (Gaussian) distribution of data were tested using D'Agostino-Pearsen omnibus normality test. A two-tailed Student's *t*-test was used to evaluate the significance of the results between two samples. For multiple comparisons tests, a one-way ANOVA followed by Tukey's multiple comparisons of means was applied. Survival curves were assessed by the log-rank (Mantel–Cox) test and conducted for each pairwise comparison. Significance levels of $P < 0.05$ (*), $P < 0.01$ (**), $P < 0.001$ (***) and $P < 0.0001$ (****) were used in all tests. The statistical tests were performed using Microsoft Excel v16.50 (Microsoft, US) and the data analysis program GraphPad Prism 9.1 (GraphPad Software Inc., CA, USA).

**Reporting summary**. Further information on research design is available in the Nature Research Reporting Summary linked to this article.

## Data availability

The raw data that support the findings of this study are available from the corresponding author upon reasonable requests. Source data are provided with this paper.

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

## Acknowledgements

We are grateful to Manfred Frasch, Achim Paululat, Diogo Manuel, Jae Park, Jan Veenstra, Dalibor Kodrík, Bloomington *Drosophila* stock Center, Vienna *Drosophila* Resource Center and Developmental Studies Hybridoma Bank for the generous sharing of resources, as well as to Michael Texada for giving critical comments to the manuscript. Camilla Trang Vo and Christina Papamichail are thanked for helping with pupal

weight and developmental timing quantifications and Manal Merimi is thanked for help with qRT-PCR analysis. This work was supported by funding from the Villum Foundation (15365) and Danish Council for Independent Research Natural Sciences (9064-00009B) to KVH. Additional funding was given by UKRI BBSRC (BB/P008097/1) to SD, JATD and ST as well as by Novo Nordisk Foundation (16OC0021270) to KR. ST was additionally funded by United Kingdom Medical Research Council (MC_UU_12014/8).

## Author contributions

T.K., S.T. and K.V.H. designed and conceptualized the study. T.K., S.T., M.T.N., S.N. and K.V.H. performed the experiments and analyzed the data. K.V.H. wrote the manuscript and produced the figures, with input from T.K. and S.T. T.K., S.T., K.R., J.A.T.D., S.A.D. and K.V.H. reviewed and edited the manuscript. K.V.H. directed the project and provided the core funding.

## Competing interests

The authors declare no competing interests
