## [Peer Review File · Nature Communications]

A nutrient-responsive hormonal circuit mediates an inter-tissue program regulating metabolic homeostasis in adult *Drosophila*Peer Review File

Reviewer comments, first round –

Reviewer #1 (Remarks to the Author):

In this manuscript, Halberg and colleagues reported a comprehensive analysis of *Drosophila* Capa/CapaR signaling in the regulation of energy homeostasis. The authors identified gut visceral muscle as one of the main target tissues of Capa/CapaR signaling. They show that visceral muscle-specific knockdown (or knockout) of CapaR decreased gut motility, fluid and waste excretion, copper cell acidity, nutrient breakdown and absorption, and food intake. Interestingly, gut visceral muscle-specific silencing of CapaR impaired locomotor activity and altered thoracic muscle physiology, including depletion of glycogen storage. In addition, they found the activities of Capa-positive Va neurons were regulated by starvation and desiccation, suggesting an important role of Capa signaling in sensing nutrient and water. Finally, the Capa/CapaR signaling was found to control AKH production in APC cells. The study has established a great model system to study the physiological roles of Capa signaling using *Drosophila* and will contribute to our understanding of the evolutionarily conserved functions between Capa and neuromedin U signaling. I have several concerns listed below that need to be addressed before the manuscript is acceptable for publication.

1. In Figure 2b-c, Capa-1-F was shown to bind to CapaR in both visceral muscles and EE cells. But EE-specific markers was not used in this experiment. Therefore, the cell identity is inclusive. It is interesting that CapaR only expressed in the visceral muscle of the midgut (nearby copper-cell region). Could the authors discuss the significance of this spatially restricted expression pattern and the potential transcriptional regulation of the CapaR expression?
2. The authors did not find any direct innervation of the intestine by Capa-positive neurons and they propose that Capa peptides could be released into circulation to control gut function. To prove this, it would be great if they can show the detection of Capa peptides from the hemolymph samples using either Capa antibodies or by expressing a tagged-Capa.
3. w1118 was used as a wild-type control in several experiments (Figure 4, 7). However, it is not a proper control genotype in those experiments. Instead, the authors should use GAL4 drivers (or UAS) crossed to wild-type flies as what they did in other experiments.
4. When testing defecation behavior, the authors found that CapaR knockdown flies produced fewer and smaller excreta. I wonder if this is actually because of reduced food uptake (as they showed in Figure 5), rather than altered gut motility. The excreta assay might not be a good way to clearly separate the two processes.
5. The loss of acidity in the copper-cell region and reduced nutrient breakdown phenotypes are quite interesting. But the authors did not further explore the possible cause of these changes. How does CapaR signaling in visceral muscle influence the acidity of the copper cells? The authors proposed that restricted CapaR expression may be important in maintaining gut compartmentalization and optimal pH. Did they check whether the gut compartmentalization is altered in CapaR knockdown flies? Could the function and spatial organization of the copper cells be altered?
6. It is unclear why CapaR knockdown was used some experiments, but CapaR KO was used in others. The CapaR expression in CapaR KO flies was not examined. There are two predicted isoforms of CapaR in fly genome. Could the two isoforms function differently (in different tissues)? Did the author measure the tissue-specific expression patterns of these two isoforms using qRT-PCR?
7. In Figure S5, Act88F-gal4 was used to knock down CapaR in skeletal muscle and to examine the

effects of skeletal muscle-specific CapaR in locomotion and energy metabolism. Interestingly, skeletal muscle-specific knockdown of CapaR did not recapitulate the phenotypes of global muscle knockout. However, the authors did not provide evidence to show whether Act88F-gal4 can effectively drive expression in skeletal muscle, and whether CapaR expression was silenced in skeletal muscle. They should try other muscle drives (like Mef2-gal4 or MHC-gal4) to make sure the observation from Act88F-gal4 is valid.

8. In Figure 6g-h, it is unclear whether the Ca²⁺ levels were measured from skeletal muscle or gut visceral muscle. The authors need to specify this in the main text.

9. It is interesting that CapaR knockout reduced the expression of plasma membrane calcium ATPase (PMCA). I wonder if authors could show that overexpression of PMCA can rescue the locomotor defect and mortality phenotypes in CapaR knockout flies.

10. In Figure 7, the authors nicely demonstrated that the activity of Capa-positive Va neurons is regulated by nutrition and desiccation. To further show the requirement of Va neuron activation in facilitating the recovery of energy homeostasis after refeeding, the authors could compare the gut function and energy storage (TAG and glycogen) between control and Capa knockdown flies during refeeding.

11. "APC" needs to spell out at its first appearance.

12. In Figure 8, authors made an interesting observation that CapaR knockout in APCs reduced the AKH protein levels and Capa-1 decreases Ca²⁺ levels in ex vivo CC preparations. However, to demonstrate that CapaR activation in the APCs represses the release of AKH into circulation, the authors need to directly measure the AKH levels in the circulation (or conditional medium) using western blots.

Reviewer #2 (Remarks to the Author):

The manuscript by Koyama et al. includes a comprehensive and detailed characterisation of the peptidergic Va neurons in *Drosophila*. The Va neurons are a small and morphologically very distinctive group of neurosecretory cells expressing CAPA peptides, which share sequence homologies with the vertebrate NMU peptide family. The function of CAPA peptides as diuretic/anti-diuretic factors is well established since several years, in substantial part by some of the co-authors. What the authors now demonstrate here is -as it seems- central and important function of CAPA peptides in balancing homeostasis regarding carbohydrates in conjunction with diuresis and food transport in the digestive tract. This is a novel and significant finding, and adds considerably to our understanding of the regulation of homeostasis in the genetic model system *Drosophila* in general, and to the so far little understood regulation of VA neuron activity in particular. Remarkably, the manuscript provides yet another striking example of obviously conserved functions of homolog peptide signalling systems (here CAPA-PVK:Neuromedins) between flies and mammals.

The manuscript is build upon an impressive series of consecutive and sound experiments which in part also build on new transgenic lines generated in this work. The manuscript is generally very well written. This is a strong contribution that adds a new player to the field of metabolic regulation in *Drosophila*.

Perhaps not surprising regarding the large number of experiments and the complexity of the matter, I nevertheless have a few remarks and 1-2 control experiments that I suggest should be addressed during a revision process:

Major comments:

- The work includes the use of several GAL4/UAS-lines generated by the authors. While the generation of a few of these lines is described in M&M-Drosophila transgenesis, others are not (page 24, first line). It is especially important to know the make-up of the CapaR-Gal4 line. It is therefore not sufficient to state "...were previously generated in-house". Please either give references where one could look up the details, or include a description in "Drosophila transgenesis".
- page 5, middle, "The penetrance of phenotypic effects scaled with temperature..." While the temperature-dependent expression of certain GAL4-constructs is well known, it is unclear whether this is really relevant here. The temperature-dependent effect strength might at least in part also be due to metabolic rate and changing water balance, which are strongly affected by temperature. Please provide experimental evidence for your statement.
- page 8, first paragraph: I am somewhat surprised based on own experience that you seem to not have encountered severe effects during larval and pupal development. Please specifically state in the manuscript whether or not you observed such effects (even if a significant fraction of flies survived well).
- page 9, second paragraph, page 10 first paragraph: the distinction between diuretic and gut peristalsis effect of CAPA is a tricky issue as feces represent the combined outcome of both. This should not be swept under the carpet. p.9: "...significant decrease in.. waste deposits..." This could at least in part also be the result of decreased diuresis. You do it once (page 10 middle "...combined effects of...") but it should be noted already from the beginning. The effects of membrane-tethered DH44 are elegant but do not represent a strong argument in this respect as it remains unclear how effective constitutive tDH44 really is (in comparison with conditional hormonal DH44).
- page 9, second paragraph, Figure S4: To demonstrate that diuresis is not a major factor, a defecation TURD assay in *Uro>CapaR-RNAi* flies should be performed.
- page 12, line 4: "...indicating that Capa signaling plays a role in regulating feeding behavior.." This is overstated, as the effect might be quite indirect as you rightly discuss on p20. Please weaken this statement.
- p13, second paragraph: The use of Act88F-Gal4 is little appropriate to study the cause of reduced locomotor activity, as flight muscles are not relevant to locomotor activity in the Trikinetics tubes. The experiments using Act88F-Gal4 do not allow to conclude that "Capa-induced changes in skeletal muscle performance is unlikely to be a major cause...". Please weaken and take a more critical stance here. If a say leg-specific muscle-driver is available, similar experiments as for the Act88F-Gal4 driver would be helpful to solve that issue.
- Aequorin-imaging, Fig. 6: it is unclear, also from M&M, which muscles are imaged here. Please specify the preparation (gut?, legs?, thorax?) either in legend or M&M. Further: I found it first surprising that CapaR-RNAi lead to an increase in the free intracellular calcium concentration, given that Capa signalling increases Ca²⁺. Yet, your experiments regarding PMCA provide a valid and interesting explanation. Nevertheless, can you exclude that the difference in basal Ca²⁺ is not an outcome of different Ca²⁺ buffering due to differences in the amount of aequorin expressed (only one UAS-effector in control, two UAS-effectors in the RNAi experiments)?
- page 21, line 16: "...strongly suggest that the age-dependent decline in locomotor activity.." This may in fact be true and is a valid idea, but there is no experiment included that evidences this over-statement. Please weaken this sentence.
- page 21, last paragraph: "Our study suggest that the Capa neurons, not not the SEG neurons, secrete Capa..." This is not a novel finding, as it has been shown long before that the SEG neurons do not produce Capa peptides (and hence cannot secrete them, Predel et al. 2004, Wegener et al.

2006). These papers should be referenced and mentioned here.

- p. 30, feeding assays, Fig. 5 g,h: Flies were transferred to BPB for 4 hours before freezing and extraction. That is much longer than a gut passage requires (see Figure 4d,f), and thus does not represent relative consumption well (some food consumed might have left the flies already). These experiments should be repeated with an adequate BPB incubation time < gut transition time.

- Fig. 8e: How can it be explained that the AKH level in starved CapaRKO animals is low? Here, CAPA should not be physiologically released.

- Fig. 8n: Was really glucose only measured? If so, why is the glucose level lower in CapaRKO?

Minor comments:

-page 3, line 4: "However, to implement..." Unclear why "however" is used here.

-page 3, line 9: "... and releases key hormones..."

-page 3, second paragraph: "The actions of CAPA/CapaR signaling..." Most likely, the actions of CAPA signaling have stayed the same for generations also in larvae. It's rather the research that has been restricted to adult animals. Please correct

-page 4, line 5: You refer to the homologies with NmU signaling. Here, the first paper suggesting this homology for PRXamides needs to be cited along with ref. #12: Melcher, C., Bader, R., Walther, S., Simakov, O., Pankratz, M.J., 2006. Neuromedin U and its putative Drosophila homolog hugin. PLoS Biology 4, e68.

-page 5, "CapaR-RNAi": I was confused with regard which line was used (e.g. when comparing Fig 1a with Fig S1e). A clear designation that specifies the different CapaR-RNAi lines should be used throughout to avoid confusion.

- page 8, middle "...the two anterior pairs..." The morphology of the adult Va neurons has already been described in detail before. Please add the appropriate references here.

- page 11, line 8: Carbon is also a component of lipids, another important dietary component.

- page 11, line 15: "..we exposed gut-specific CapaR..." Fig. 5e shows that how-G4 was used as a driver. This line is not gut-specific, but rather muscle-specific. Compare also to p12, line 3. Please correct. Similar: p12, line 11 (..visceral muscle-specific..)

- page 16, second paragraph: "GFP reporter expression" is a bit confusing, please specify that it is a CaLexA signal.

- page 16, second paragraph: yeast as food source does not represent amino acids only, but also lipids and other components (incl. low amount of carbohydrates, which might be neglectable). Please refer to carbohydrate vs. yeast/rich instead of carbohydrate vs. amino acids, or repeat the experiments with a defined amino acid diet.

- page 17, line 12: "..detect GFP expression.." Please specify that this refers to CapaR>GFP expression.

- page 22, first line: "... results in supression of Capa..."

- page 23, fly stocks: ..CapaR-RNAi, number is 27275.

- page 26, Immunocytochemistry: please specify whether samples were treated in parallel/same

microscope settings were used for the experiments involving quantitative ICC.

- page 27 "Real-time measurements of intracellular calcium release" In your muscle experiments, calcium also comes from extracellular compartment. Please correct the heading.

- page 28, line 1: Please specify whether Schneider's contained glutamate.

- page 29: please specify laser wave length for the calcium imaging experiments. The title "Calcium imaging" is also a bit confusing, as also the aequorin experiments represent calcium imaging. I suggest to change the heading to "calcium imaging in AKH cells"

- p. 32, locomotor assay: Please give reference for pySolo.

- p33, Ramsay assay: Please specify the composition of the Drosophila saline.

- Fig. S2: legend for g) is missing.

Reviewer #3 (Remarks to the Author):

The manuscript of Koyama and colleagues describes the physiological relevance of the capa peptide capa receptor system in Drosophila, where it has similar functions as the Neuromedin U system of vertebrates. Most importantly, the authors could show formerly unknown physiological roles of capa receptor signaling in a number of different tissues. Two different arms of the capa receptor based signaling system have been identified, one focusing on the intestine and the uptake of nutrients and the other on regulation of the AKH axis. These two axes work together to control nutrient uptake and energy storage.

The manuscript describes a highly interesting topic and is of very high quality (at least in most of its parts). Nevertheless, some issues need clarification:

The most dramatic phenotype was induced by silencing CapR signaling in the intestine, more precisely in the visceral muscle of the intestine. To obtain a better understanding of this system and the multiple actions of Capa on intestinal physiology, the corresponding Capa responsive systems of the intestine have to be described in more detail. This is especially relevant for the enteroendocrine cells and for the copper cell region. Which cells are exactly Capa responsive and which peptide do they express? What is the exact target in the copper cell region? Does reducing CapaR Expression in these cells show the expected phenotype?

It is interesting that the reduction of intestinal motility is lethal after a few days. It would be interesting to know if this reduced motility or other factors induce lethality. Related to an earlier question, what is the factor that reduces uptake of nutrients – the reduced gut motility should not be decisive for this?

REVIEWER COMMENTS (NCOMMS-20-35659)

The authors are pleased with the thoughtful and helpful reviews provided by the reviewers and are furthermore convinced that they have greatly improved the quality of the manuscript. Overall, we are delighted that the reviewers have shared our excitement about the work, and we further believe that we have addressed all points raised in full. Our point-by-point response to the reviewers' critic is appended below.

Reviewer #1 (Remarks to the Author):

In this manuscript, Halberg and colleagues reported a comprehensive analysis of *Drosophila* Capa/CapaR signaling in the regulation of energy homeostasis. The authors identified gut visceral muscle as one of the main target tissues of Capa/CapaR signaling. They show that visceral muscle-specific knockdown (or knockout) of CapaR decreased gut motility, fluid and waste excretion, copper cell acidity, nutrient breakdown and absorption, and food intake. Interestingly, gut visceral muscle-specific silencing of CapaR impaired locomotor activity and altered thoracic muscle physiology, including depletion of glycogen storage. In addition, they found the activities of Capa-positive Va neurons were regulated by starvation and desiccation, suggesting an important role of Capa signaling in sensing nutrient and water. Finally, the Capa/CapaR signaling was found to control AKH production in APC cells. The study has established a great model system to study the physiological roles of Capa signaling using *Drosophila* and will contribute to our understanding of the evolutionarily conserved functions between Capa and neuromedin U signaling. I have several concerns listed below that need to be addressed before the manuscript is acceptable for publication.

1. In Figure 2b-c, Capa-1-F was shown to bind to CapaR in both visceral muscles and EE cells. But EE-specific markers was not used in this experiment. Therefore, the cell identity is inclusive.

The CapaR expressing cells in the midgut, which specifically bind the Capa-1-F ligand, have the characteristic shape of an EE secretory cell as shown in Fig. 2b, while CapaR expression in the EEs and VMs was confirmed by FACS sorted cell populations as shown in FigS2h. However, we agree with the reviewer that, to unambiguously demonstrate the identity of the cell type expressing CapaR, we counterstained *CapaR>mCD8::GFP* flies with an antibody directed against the EE marker, Prospero, as well as a combination of antisera recognizing different peptide hormones (Tachykinin, DH31, CCHa1, AstC and NPF) known to be expressed and released from EEs. These data not only verify the cellular identity of the CapaR expressing EEs (all CapaR⁺ cells are Prospero positive, but all Prospero positive cells are not CapaR⁺), but also reveal which combination of hormones these cells additionally express. These data have been included in Fig. S2i and mentioned in the main text (page 7, first paragraph) in the revised manuscript.

It is interesting that CapaR only expressed in the visceral muscle of the midgut (nearby copper-cell region). Could the authors discuss the significance of this spatially restricted expression pattern and the potential transcriptional regulation of the CapaR expression?

We, like the reviewer, found it very intriguing that CapaR expression was restricted to distinct portions of the gut containing valves and sphincters, including the proventriculus and the rectum as well as to the region flanking the midgut acidic zone. We have initially discussed these points in the original manuscript (page 21, second paragraph and page 22, second paragraph) and now expanded this slightly in the discussion according to the reviewer's suggestion.

2. The authors did not find any direct innervation of the intestine by Capa-positive neurons and they propose that Capa peptides could be released into circulation to control gut function. To prove this, it would be great if they can show the detection of Capa peptides from the hemolymph samples using either Capa antibodies or by expressing a tagged-Capa.

The direct detection and accurate quantification of small circulating peptides, such as Capa peptides (10-12 aa long), in *Drosophila* hemolymph is notoriously difficult and not a straightforward task; a point recently emphasized by MacMillan et al.¹. To our knowledge, only larger protein hormones, such as Dilp2 and Bursa^{2,3}, have been directly detected in *Drosophila* hemolymph samples, given that their large size enables their detection using western blotting. Other protein hormones, such as FIT⁴, have been detected by overexpressing tagged-proteins, yet it remains highly questionable whether such analyses reflect physiological conditions. In principle, one could potentially detect overexpressed Capa peptides by ELISA if specific antibodies recognizing capa-1 and/or capa-2 existed, but the only anti-Capa antibody available is directed against a linking region of the prepropeptide that does not encode any biologically active neuropeptide. In addition, since the Capa peptides belong to the larger PRXamide family of peptides with similar COOH-terminals (1. the Pyrokinin or PBAN-like peptides defined by the C-terminal motif FXPRXamide, 2. the periviscerokinin or Cap2b-like peptides ending in FPRXamide and 3. the ETH-like peptides with PRXamide sequence motif), it is unlikely that we can specifically detect mature Capa signals using antibody-based strategies without experiencing cross-reactivity issues (see e.g. Kean et al.⁵)

Our neuroanatomical analysis reveals strong anti-Capa immunoreactivity in the neurohemal release sites innervated by the Capa producing Va neurons. These neurohemal organs are known to store and release hormones into the hemolymph as e.g. evidenced by direct mass spectrometric profiling data, showing that Capa hormones are the most abundant peptides in these neurohemal sites^{6,7}, as well as by the conspicuous fusion profiles of immunogold-labelled Capa⁺ vesicles⁸. Furthermore, the systemic release of Capa hormones has also been directly demonstrated for other higher insects⁹. These points have been included in the revised manuscript on page 8, second paragraph.

However, to address the point raised by the reviewer, while circumventing the abovementioned experimental difficulties or absence of reagents, we developed an alternative strategy, in which we transplanted CC from *Akh^{ts}>CapaR^{KO}* or control donor flies into host animals expressing the heat-sensitive K⁺ channel, TRPA1, in Capa⁺ neurons, in order to generate heat-induced Capa titers under *in vivo*-like conditions (*Trp>TRPA1*). After incubating the transplanted flies at the inducible temperature (30°C) we recovered the freshly dissected control and *CapaR* mutant transplanted CC and probed them for anti-AKH activity. Given that the dissected CC were blocked from

neuronal inputs, differences in intracellular AKH levels in CapaR knockout relative to control CC is most likely explained by Capa hormone-induced changes in AKH release. These data show that the AKH-producing cells (APCs) from *Akh^{ts}*>+ flies retain significantly more AKH compared to untransplanted controls, whereas the AKH levels in the APCs of *Akh^{ts}*>*CapaR^{KO}* flies remained significantly lower than controls in both conditions. Together, these data support a model in which Capa peptides are secreted into the circulating hemolymph to act on the APCs to reduce AKH release. We thus believe that this experiment has successfully addressed the reviewers concern, as we have, albeit indirectly, shown that Capa peptides are released in to circulation wherefrom they control several physiological processes, including AKH secretion. The data is included in Figure 8j-k and described in the revised manuscript (page 19, last paragraph).

3. w1118 was used as a wild-type control in several experiments (Figure 4, 7). However, it is not a proper control genotype in those experiments. Instead, the authors should use GAL4 drivers (or UAS) crossed to wild-type flies as what they did in other experiments.

Throughout this study, we have been extremely careful in comparing the phenotypes observed in CapaR KD/KO animals with both parental controls that have been outcrossed to produce heterozygotic animals. However, the experiments referred to in Figures 4 and 7 in which we used *w¹¹¹⁸* flies, are not meant to represent controls to the experiments in which we carry out genetic manipulations. Rather, these are independent experiments designed to explore the role of Capa signaling in the most suitable genetic background (the white null background was used for all our transgenics) with which to carry out these experiments. In Figure 4, we tested how Capa peptide application changes gut contraction frequency relative to unstimulated guts and is therefore a paired experiment in which unstimulated guts act as controls. To make it clearer that they constitute separate experiments, we have split the panels (a to a + b and b to c + d) so as to avoid ambiguity. Similarly, in Figure 7 we assessed whether Capa signaling activity is augmented by environmental stress conditions in *w¹¹¹⁸* flies, which is an independent experiment to that in which we use *Trp>CaLexA* flies to quantify Va neuron activity during identical conditions. The experiments in the two figures are thus separate, but complementary, which is further illustrated by the fact that we do not test for statistical differences between the two data sets. To address the reviewer's concerns, we have tried to make these points clearer in the revised manuscript, by emphasizing that *w¹¹¹⁸* refers to the most suitable genetic background with which to carry out these experiments, and not direct controls to the experiments where we used transgenics, in the appropriate results sections (page 9, second paragraph).

4. When testing defecation behavior, the authors found that CapaR knockdown flies produced fewer and smaller excreta. I wonder if this is actually because of reduced food uptake (as they showed in Figure 5), rather than altered gut motility. The excreta assay might not be a good way to clearly separate the two processes.

We agree with the point raised by the reviewer; we cannot fully exclude that lower food intake also contributes to a lower defecation rate observed in *how^{ts}*>*CapaR^{RNAi}* flies. However, using this assay we discovered additional differences in fecal output

resulting from *CapaR* knockdown, such as the production of smaller excreta with a higher dye intensity. These effects are uncoupled from the amount of food ingested, but are instead consistent with changes in intestinal fluid contents¹⁰, thus pointing to gut-induced changes in fluid excretion in *CapaR* depleted animals. The excreta assay have therefore provided us with additional information on the physiological consequences of reduced gut motility *in vivo* – *i.e.* slower gut transit time impacts intestinal fluid balance – which incidentally complements our intestinal motility and gut transit time experiments.

5. The loss of acidity in the copper-cell region and reduced nutrient breakdown phenotypes are quite interesting. But the authors did not further explore the possible cause of these changes. How does *CapaR* signaling in visceral muscle influence the acidity of the copper cells? The authors proposed that restricted *CapaR* expression may be important in maintaining gut compartmentalization and optimal pH. Did they check whether the gut compartmentalization is altered in *CapaR* knockdown flies? Could the function and spatial organization of the copper cells be altered?

This is a very good point raised by the reviewer. We discuss a possible mechanism with which *Capa/CapaR* signaling might affect midgut acidity in the discussion (second paragraph) of the original manuscript. Yet, to further explore the causal mechanisms underpinning the reduced nutrient digestion and absorption phenotype of *How^{ts}>CapaR^{KO}* flies, we have performed new experiments to test whether acute reduction in CCR acidity is necessary and sufficient to reduce nutrient absorption in adult animals (the loss of CCR acidity and therefore loss of gut compartmentalization was also observed in *how^{ts}>CapaR^{RNAi}* animals). To do this, we impaired CCR acid secretion by knocking down a subunit of the Vacuolar H⁺-ATPase (*Vha55*), a membrane transporter enriched in the CCR and critical for midgut acid generation¹¹, using a CCR-specific GAL4 driver line¹² (*Lab^{ts}>Vha55-RNAi*), and measured the resulting change in nutrient digestion and absorption by quantifying the amount of undigested nutrients in fly excreta. These data show that targeted knockdown of *Vha55* expression effectively neutralizes CCR acidity and that the amount of undigested nutrients is significantly increased relative to control and that this induces an increased mortality in these flies. Similar reductions in organismal lifespan has previously been reported in flies in which the CCR was abolished by expressing *Lab^{RNAi}*¹³. These results indicate that acute disruption of CCR acid secretion is sufficient to impair nutrient digestion and absorption in adult flies, and therefore, at least partly, explain – in combination with reduced gut motility – the observed metabolic profile of *how^{ts}>CapaR^{KO}* flies. These data are now included in Fig. S5 and described on page 12, first paragraph.

6. It is unclear why *CapaR* knockdown was used some experiments, but *CapaR* KO was used in others. The *CapaR* expression in *CapaR* KO flies was not examined.

There are two predicted isoforms of *CapaR* in fly genome. Could the two isoforms function differently (in different tissues)? Did the author measure the tissue-specific expression patterns of these two isoforms using qRT-PCR?

The RNAi knockdown and CRISPR/Cas9 knockout constructs were used to demonstrate that the two independent genetic tools consistently showed similar results; in all experiments performed, both strategies produced comparable data, albeit with *CapaR* KO flies displaying stronger phenotypic effects as expected. Indeed, several of the

phenotypes were intentionally investigated using both types of genetic manipulations (e.g., Fig. 1, Fig. 5 and Fig. 6) in order to emphasize this point and to validate our observations. As suggested by the reviewer, the silencing efficacy of our CRISPR/Cas9 knockout construct was validated by qRT-PCR and is now shown in Fig.S1e and mentioned on page 5, first paragraph.

A concept of two functionally different CapaR isoforms mediating different tissue-specific actions is attractive. However, the presence of a second CapaR-RC isoform must be regarded as a highly questionable annotation, given that the sequence of this predicted isoform is identical to the -RB isoform, except for a premature stop codon, which results in a truncated protein that only contains five transmembrane domains due to the elimination of aa 285-457 of the C-terminal part of the protein. This missing region is known to be critical for proper GPCR:G-protein engagement and receptor activation in Class A GPCRs¹⁴, which strongly suggests that this isoform is non-functional. Additionally, only one Capa receptor protein is identified/predicted in other all dipteran genomes sequenced to date. We have therefore removed the exon map of -RC isoform from the Fig.S1a to avoid any further confusion.

7. In Figure S5, Act88F-gal4 was used to knock down CapaR in skeletal muscle and to examine the effects of skeletal muscle-specific CapaR in locomotion and energy metabolism. Interestingly, skeletal muscle-specific knockdown of CapaR did not recapitulate the phenotypes of global muscle knockout. However, the authors did not provide evidence to show whether Act88F-gal4 can effectively drive expression in skeletal muscle, and whether CapaR expression was silenced in skeletal muscle. They should try other muscle drives (like Mef2-gal4 or MHC-gal4) to make sure the observation from Act88F-gal4 is valid.

The reviewer raises several important points, which we have now addressed both in the manuscript and discussed below:

Bryantsev et al. (2012) has previously shown that *Act88F-GAL4* effectively drives expression in the indirect flight muscles and this was cited in the original manuscript¹⁵. This observation was also concluded by Nongthomba et al. (2001) – now also referenced in the revised manuscript (page 14, second paragraph) – who further showed that in addition to the flight muscles, *Act88F-Gal4* also drives expression in the leg and abdominal muscles¹⁶. We have now also verified these reports by looking at *Act88F-Gal4* driven *mCD8::GFP* expression, confirming that GFP is indeed expressed in flight and leg muscles but also in non-overlapping visceral muscles of the anterior midgut; these data have been included in Fig. S6a-c and mentioned on page 14, second paragraph. Thus, *Act88F-Gal4* exclusively drives expression in the flight, leg and abdominal muscles, but not in *CapaR*-expressing visceral muscles, of adult flies.

To assess the efficacy of *CapaR* knockout using the *Act88F* driver as suggested by the reviewer, we performed qRT-PCR analysis on thoraces of *Act88F>CapaR^{KO}* flies, showing that *CapaR* expression is almost completely abolished in these animals. These results are also shown in Fig. S6d and mentioned on page 14, second paragraph in the revised manuscript.

We thank the reviewer for the suggestion of using either *Mef2-* or *MHC-Gal4* lines to independently verify the data obtained using *Act88F-Gal4* driver line. However, both of these Gal4 lines drive expression in all muscles¹⁷, similar to *how-Gal4*, and are

therefore unsuitable for determining the exact contribution of skeletal and visceral muscles to the observed phenotypes following CapaR KD/KO.

8. In Figure 6g-h, it is unclear whether the Ca²⁺ levels were measured from skeletal muscle or gut visceral muscle. The authors need to specify this in the main text. Fig. 6f-I all shows data relating to skeletal muscles as we have indicated in the figure. To avoid any confusion, we have specified this further on page 15, second paragraph as well as in the appropriate figure legend as suggested by the reviewer.

9. It is interesting that CapaR knockout reduced the expression of plasma membrane calcium ATPase (PMCA). I wonder if authors could show that overexpression of PMCA can rescue the locomotor defect and mortality phenotypes in CapaR knockout flies. This is an interesting question that the authors are currently considering in a subsequent study, as it requires the generation of a UAS-PMCA fly line, which isn't publicly available. However, given that our data suggest that reduced PMCA activity and Ca²⁺-dysregulation is mainly caused by a depletion of internal energy stores, rather than major defects in the Ca²⁺-extrusion mechanisms *per se*, we hypothesize that overexpression of a PMCA construct is unlikely to rescue the observed locomotor and mortality defects.

10. In Figure 7, the authors nicely demonstrated that the activity of Capa-positive Va neurons is regulated by nutrition and desiccation. To further show the requirement of Va neuron activation in facilitating the recovery of energy homeostasis after refeeding, the authors could compare the gut function and energy storage (TAG and glycogen) between control and Capa knockdown flies during refeeding.

To address this valid point raised by the reviewer, we generate a *Trp^{ts}* line and used it to knock down *Capa* expression specifically in the adult stage and subsequently measured whole-animal energy levels. These data show that the recovery of energy storage levels (glucose, TAG and glycogen levels) is significantly reduced in *Trp^{ts}>Capa^{RNAi}* animals relative to controls, thus phenocopying the effects observed in muscle-specific *CapaR* knockout flies. These data, along with qRT-PCR validation of Capa knockdown as well as the long-term survival of the *Trp^{ts}>Capa^{RNAi}* flies, have been added as Fig. 7e-f as well as Fig.S7d and were described on page 17, last paragraph.

11. "APC" needs to spell out at its first appearance.

APC was defined at first use on page 8, first paragraph in the original manuscript.

12. In Figure 8, authors made an interesting observation that CapaR knockout in APCs reduced the AKH protein levels and Capa-1 decreases Ca²⁺ levels in ex vivo CC preparations. However, to demonstrate that CapaR activation in the APCs represses the release of AKH into circulation, the authors need to directly measure the AKH levels in the circulation (or conditional medium) using western blots.

As previously argued, the robust quantification of small circulating peptides, such as AKH (8 aa long), is an exceedingly difficult task in *Drosophila*. Yet, to directly address the reviewer's suggestion, we attempted to detect changes in hemolymph AKH levels by using a modified dot-blot assay (western blotting is in our experience not suitable to detect peptides as small as AKH). To do this, we harvested hemolymph from 800

flies from both control ($Akh^{ts}>+$) and knockout flies ($Akh^{ts}>CapaR^{KO}$) and measured hemolymph AKH levels relative to that of dot-blot of known AKH concentrations. These data revealed that $Akh^{ts}>CapaR^{KO}$ flies contained significantly more AKH in circulation relative to that of control flies, consistent with a model in which Capa signaling acts to repress AKH release from the APCs. Moreover, we independently addressed this question using a genetic approach. To do this we co-overexpressed $Akh-RNAi$ with $CapaR^{KO}$, which according to our hypothesis should rescue the observed starvation hypersensitivity. In line with this notion, the results show that co-expressing $Akh-RNAi$ with $CapaR^{KO}$ in the APCs significantly increased starvation resistance to a level in which survival was not different from that of flies carrying the $Akh-RNAi$ construct alone. These additional data are now included as Figure 8d,g; Figure S8a and described in the main text on page 19, first and second paragraph.

Reviewer #2 (Remarks to the Author):

The manuscript by Koyama et al. includes a comprehensive and detailed characterisation of the peptidergic Va neurons in *Drosophila*. The Va neurons are a small and morphologically very distinctive group of neurosecretory cells expressing CAPA peptides, which share sequence homologies with the vertebrate NMU peptide family. The function of CAPA peptides as diuretic/anti-diuretic factors is well established since several years, in substantial part by some of the co-authors. What the authors now demonstrate here is -as it seems- central and important function of CAPA peptides in balancing homeostasis regarding carbohydrates in conjunction with diuresis and food transport in the digestive tract. This is a novel and significant finding, and adds considerably to our understanding of the regulation of homeostasis in the genetic model system *Drosophila* in general, and to the so far little understood regulation of VA neuron activity in particular. Remarkably, the manuscript provides yet another striking example of obviously conserved functions of homolog peptide signalling systems (here CAPA-PVK:Neuromedins) between flies and mammals.

The manuscript is build upon an impressive series of consecutive and sound experiments which in part also build on new transgenic lines generated in this work. The manuscript is generally very well written. This is a strong contribution that adds a new player to the field of metabolic regulation in *Drosophila*.

Perhaps not surprising regarding the large number of experiments and the complexity of the matter, I nevertheless have a few remarks and 1-2 control experiments that I suggest should be addressed during a revision process:

Major comments:

- The work includes the use of several GAL4/UAS-lines generated by the authors. While the generation of a few of these lines is described in M&M-*Drosophila* transgenesis, others are not (page 24, first line). It is especially important to know the make-up of the CapaR-Gal4 line. It is therefore not sufficient to state "...were previously generated in-house". Please either give references where one could look up the details, or include a description in "*Drosophila* transgenesis".

We thank the reviewer for pointing this out. Although the references providing the details for these previously generated fly lines were included elsewhere in the original

manuscript, we have now also listed them at their appropriate places in the M&M on page 25, second paragraph in the revised manuscript for more clarity.

- page 5, middle, "The penetrance of phenotypic effects scaled with temperature..." While the temperature-dependent expression of certain GAL4-constructs is well known, it is unclear whether this is really relevant here. The temperature-dependent effect strength might at least in part also be due to metabolic rate and changing water balance, which are strongly affected by temperature. Please provide experimental evidence for your statement.

The rationale behind this experiment was to leverage the temperature-dependent effects of the GAL4/UAS-system to obtain animals with different degrees of *CapaR* knockdown, since we initially wanted to evaluate the effects of silencing *CapaR* on larval development and adult survival. At lower temperatures, we found that the severity of the adult survival phenotype was significantly reduced relative to that of higher temperatures. To test whether this effect correlates with *CapaR* knockdown efficacy, we quantified *CapaR* expression using qRT-PCR in animals globally expressing the *UAS-CapaR^{RNAi}* construct at three different temperatures. These data show that *CapaR* knockdown scales with temperature, thus supporting our claim - the results have now been presented as Fig. S1b in the revised manuscript. However, we agree with the point raised by the reviewer; even though we assess both parental lines alongside the experimental animals, we cannot exclude that the genetic effects are synergized by environmental factors. Accordingly, we have toned down our statement on page 5, second paragraph, consistent with the reviewer's suggestion.

- page 8, first paragraph: I am somewhat surprised based on own experience that you seem to not have encountered severe effects during larval and pupal development. Please specifically state in the manuscript whether or not you observed such effects (even if a significant fraction of flies survived well).

In our hands, knocking down *CapaR* expression did not induce noticeable lethality during larval development. Indeed, we failed to detect any defects in pupal size or developmental timing, which are generally accepted as 'gold-standards' for assessing *Drosophila* developmental defects^{18,19}; these data were presented as Figure S1c-d in the original manuscript. We are currently planning a follow up study to look more closely at the effects of *Capa/CapaR* depletion during larval development.

In contrast, when knocking down *Capa* gene expression or specifically ablating the *Capa* neurons using *reaper* transgene, we found significant L2 or L3 lethality and a marked reduction in the number of eclosed adults when raised at 25°C. The *Capa* gene encodes three neuropeptides: *Capa*-periviscerokinin-1 and -2 (*Capa*-PVK-1 and -2) and *Capa*-pyrokinin-1 (*Capa*-PK-1), a member of the pyrokinin family of peptides exerting distinct functions to that of the *Capa*-PVKs. Since the Va neurons produce and release all *Capa* products⁶, these developmental defects could be associated with the elimination of *Capa*-PK-1 signaling. Interestingly, similar eclosion phenotype was found upon manipulation of the *Drosophila hugin* gene which encodes for *Hugin*-PK (pyrokinin-2; PK-2) and ecdysis-triggering hormone (ETH) neuropeptides. As suggested by the reviewer, we have specifically stated this observation in the main text (page 8, middle page) of the revised manuscript.

- page 9, second paragraph, page 10 first paragraph: the distinction between diuretic and gut peristalsis effect of CAPA is a tricky issue as feces represent the combined outcome of both. This should not be swept under the carpet. p.9: "...significant decrease in.. waste deposits..." This could at least in part also be the result of decreased diuresis. You do it once (page 10 middle "...combined effects of...") but it should be noted already from the beginning. The effects of membrane-tethered DH44 are elegant but do not represent a strong argument in this respect as it remains unclear how effective constitutive tDH44 really is (in comparison with conditional hormonal DH44).

These are good recommendations and we have now specified in the beginning of this paragraph that the excretory material consist of the combined activities of both the gut and the renal tubules excretory and osmoregulatory systems. Moreover, we have included supporting data in Fig. S4a showing that overexpressing of *UAS-tDh44* in tubule principal cells (*Uro>tDh44*) results in a significantly elevated basal rate of fluid secretion relative to parental controls demonstrating the effectiveness of constitutive activation of the DH44 pathway using membrane-tethered DH44. These data are now mentioned on page 11, first paragraph.

- page 9, second paragraph, Figure S4: To demonstrate that diuresis is not a major factor, a defecation TURD assay in *Uro>CapaR-RNAi* flies should be performed.

We agree with the reviewer and we have additionally analyzed the defecation behavior of *Uro>CapaR-RNAi* flies. These data are presented in Fig. S4d and reveal that *CapaR* silenced flies do produce fewer deposits, yet with similar size and dye intensity relative to controls. These changes in the fecal output profile, however, similarly fail to phenocopy the full effects observed in *how^{ts}>CapaR^{RNAi}* flies (producing fewer, smaller and more concentrated deposits), confirming that gut peristalsis significantly impacts fecal excretion. These data are similarly mentioned on page 11, first paragraph of the revised manuscript.

- page 12, line 4: "...indicating that *Capa* signaling plays a role in regulating feeding behavior.." This is overstated, as the effect might be quite indirect as you rightly discuss on p20. Please weaken this statement.

We have weakened the statement as suggested, now stating: "...may modulate feeding behavior either directly or indirectly.."

- p13, second paragraph: The use of *Act88F-Gal4* is little appropriate to study the cause of reduced locomotor activity, as flight muscles are not relevant to locomotor activity in the Trikinetics tubes. The experiments using *Act88F-Gal4* do not allow to conclude that "*Capa*-induced changes in skeletal muscle performance is unlikely to be a major cause...". Please weaken and take a more critical stance here. If a say leg-specific muscle-driver is available, similar experiments as for the *Act88F-Gal4* driver would be helpful to solve that issue.

We thank the reviewer for the helpful suggestion regarding the use of *Act88F-Gal4* driver. As explained to the reviewer 1 - comment 7, we think that there is no need of using another driver as it was previously shown that *Act88F-Gal4* also drives expression in the leg and abdominal muscles¹⁶ (now referenced page 14, second paragraph); data are included as Fig. S6a-c (*Act88F-Gal4* driven *mCD8::GFP* validation)

and Fig. S6d (qRT-PCR analysis of *Act88F>CapaR^{KO}* flies) and mentioned on page 14, second paragraph in the revised manuscript.

- Aequorin-imaging, Fig. 6: it is unclear, also from M&M, which muscles are imaged here. Please specify the preparation (gut?, legs?, thorax?) either in legend or M&M. Further: I found it first surprising that *CapaR*-RNAi lead to an increase in the free intracellular calcium concentration, given that *Capa* signalling increases Ca^{2+} . Yet, your experiments regarding *PMCA* provide a valid and interesting explanation. Nevertheless, can you exclude that the difference in basal Ca^{2+} is not an outcome of different Ca^{2+} buffering due to differences in the amount of aequorin expressed (only one UAS-effector in control, two UAS-effectors in the RNAi experiments)?

Thank you for pointing this out to us. We have now specified which muscles were sampled in both the figure legend and in the M&M. For clarification, the calcium reporter is based on a translational fusion of GFP and apoaequorin which have been shown to greatly increased stability and luminescence²⁰, allowing superior real-time recordings to be obtained with less tissue in each sample²¹. Therefore, our calcium measurement is not based using an imaging reporter but rather as absolute measurement of intracellular calcium. In our aequorin-based Ca^{2+} -measurements, only one copy of apoaequorin is used in either *how,GFP::aeq>CapaR^{RNAi}*; *how,GFP::aeq>PMCA^{RNAi}* and control *how,GFP::aeq/+* samples. In addition, the total amount of aequorin used and remaining during the experiment is known for each sample which allow us to accurately “normalize” and determine $[Ca^{2+}]_i$, as calculated by backward integration. Taken together, the basal Ca^{2+} increase measured in both *CapaR* and *PMCA* knockdown midgut tissue is a strong phenotype which would be interesting to further investigate.

- page 21, line 16: “.strongly suggest that the age-dependent decline in locomotor activity..” This may in fact be true and is a valid idea, but there is no experiment included that evidences this over-statement. Please weaken this sentence.
We have weakened this statement as suggested, now saying “may imply” rather than “strongly suggest”

- page 21, last paragraph: “Our study suggest that the *Capa* neurons, not not the SEG neurons, secrete *Capa*...” This is not a novel finding, as it has been shown long before that the SEG neurons do not produce *Capa* peptides (and hence cannot secrete them, Predel et al. 2004, Wegener et al. 2006). These papers should be referenced and mentioned here.

We are happy to do this and have now referenced both papers at the suggested place in the manuscript.

- p. 30, feeding assays, Fig. 5 g,h: Flies were transferred to BPB for 4 hours before freezing and extraction. That is much longer than a gut passage requires (see Figure 4d,f), and thus does not represent relative consumption well (some food consumed might have left the flies already). These experiments should be repeated with an adequate BPB incubation time < gut transition time.

We have repeated these experiments using 30-min-fed adults instead as suggested. These data are presented in Figure 5g-h of the revised manuscript and show the same hypophagic response by *Capa/CapaR* deficient animals as shown before.

- Fig. 8e: How can it be explained that the AKH level in starved *CapaRKO* animals is low? Here, CAPA should not be physiologically released.

The increase in AKH secretion upon *CapaR* elimination suggests that the Va neurons also release *Capa* peptides, albeit at lower levels, during starvation. This observation is consistent with the quantifications of *Capa* immunoreactivity and CaLexA-induced GFP intensities, which show that the *Capa*⁺ Va neurons become less active, but are not completely inactive, during periods of nutrient deprivation (see Fig. 7c-d). This effect is furthermore in line with the observed hypersensitivity to starvation and the reduced levels of metabolites exhibited by *Akh>CapaR^{KO}* flies relative to control. These results imply that although we believe that the main role of *Capa/CapaR* signaling is to repress AKH release during feeding to avoid harmful hyperglycemia, our data also suggest that *Capa* peptides play a role in restricting AKH release during starvation to potentially prevent overuse of internal energy reserves. We have now added a comment to reflect this fact on page 20, first paragraph.

- Fig. 8n: Was really glucose only measured? If so, why is the glucose level lower in *CapaRKO*?

We did not only measure glucose but additionally quantified TAG, glycogen and trehalose levels in these animals, which have now been included in Fig. 8o-r of the revised manuscript. These data similarly show a reduction in organismal energy levels, which we believe can be explained by the combined effects of nutrient deprivation and increased mobilization of nutrient stores; effects that collectively reduce starvation resistance in these animals. Indeed, the flies used in the "starved" group were sampled after 24 hours of starvation, at which time we observed almost 25% mortality (Fig. 8c). We therefore do not believe it is overly surprising that *Akh^{ts}>CapaR^{KO}* flies possess lower glycemic levels and internal energy stores, since these flies have simply depleted their energy reserves at this point. We have also included a comment on page 20, first paragraph to further explain this point.

Minor comments:

-page 3, line 4: "However, to implement..." Unclear why "however" is used here. We have removed "However" from the sentence.

-page 3, line 9: "... and releases key hormones..." Changed.

-page 3, second paragraph: "The actions of CAPA/*CapaR* signaling..." Most likely, the actions of CAPA signaling have stayed the same for generations also in larvae. It's rather the research that has been restricted to adult animals. Please correct Corrected.

-page 4, line 5: You refer to the homologies with NmU signaling. Here, the first paper suggesting this homology for PRXamides needs to be cited along with ref. #12: Melcher, C., Bader, R., Walther, S., Simakov, O., Pankratz, M.J., 2006. Neuromedin U and its putative Drosophila homolog hugin. PLoS Biology 4, e68.

We have included this reference in the main text.

-page 5, "CapaR-RNAi": I was confused with regard which line was used (e.g. when comparing Fig 1a with Fig S1e). A clear designation that specifies the different CapaR-RNAi lines should be used throughout to avoid confusion.

The most penetrant RNAi phenotype is produced by the in-house generated doubly homozygous line (chromosome II + III, see reference 12 in the original manuscript), the only one used in the main manuscript, and is referred to as *CapaR^{RNAi}* in the figure (explained as 2xRNAi in the legend), whereas the two other independent RNAi-lines are named #2 and #3, respectively, and used only in Fig. S1f in the original manuscript. This should hopefully avoid any confusion as to which RNAi lines were used.

- page 8, middle "...the two anterior pairs..." The morphology of the adult Va neurons has already been described in detail before. Please add the appropriate references here.

We completely agree and have included appropriate references here.

- page 11, line 8: Carbon is also a component of lipids, another important dietary component.

We thank the reviewer for bringing this to our attention. We have now also mentioned lipids in the revised manuscript.

- page 11, line 15: "...we exposed gut-specific CapaR..." Fig. 5e shows that how-G4 was used as a driver. This line is not gut-specific, but rather muscle-specific. Compare also to p12, line 3. Please correct. Similar: p12, line 11 (...visceral muscle-specific..)

This is a valid point and we now have changed "visceral" and "gut-specific" to "muscle-specific" where appropriate.

- page 16, second paragraph: "GFP reporter expression" is a bit confusing, please specify that it is a CaLexA signal.

As suggested, we have now changed to "CaLexA-GFP signal".

- page 16, second paragraph: yeast as food source does not represent amino acids only, but also lipids and other components (incl. low amount of carbohydrates, which might be neglectable). Please refer to carbohydrate vs. yeast/rich instead of carbohydrate vs. amino acids, or repeat the experiments with a defined amino acid diet.

We thank the reviewer for the helpful suggestion and we have changed "amino acids" to "yeast" in this paragraph.

- page 17, line 12: "...detect GFP expression..." Please specify that this refers to CapaR>GFP expression.

Done.

- page 22, first line: "... results in suppression of Capa..."
Done.

- page 23, fly stocks: ..CapaR-RNAi, number is 27275.
Corrected.

- page 26, Immunocytochemistry: please specify whether samples were treated in parallel/same microscope settings were used for the experiments involving quantitative ICC.
All quantitative estimates of intracellular hormone levels were performed on z-stacks acquired using identical microscope settings (within each experiment) as now specified on page 28, second paragraph.

- page 27 "Real-time measurements of intracellular calcium release" In your muscle experiments, calcium also comes from extracellular compartment. Please correct the heading.
We have changed the heading to "...intracellular calcium changes" to avoid specifying where Ca^{2+} is mobilized from.

- page 28, line 1: Please specify whether Schneider's contained glutamate.
The Schneider's *Drosophila* medium used for these experiments contains glutamate (L-glutamic acid). We refer to the distributor (ThermoFisher Scientific) for the complete formulation. This is different from the "HL3.1 AHL" solution²² used for the live Ca^{2+} -imaging, and which does not contain glutamate.

- page 29: please specify laser wave length for the calcium imaging experiments. The title "Calcium imaging" is also a bit confusing, as also the aequorin experiments represent calcium imaging. I suggest to change the heading to "calcium imaging in AKH cells"
We have made corrections according to the reviewer's suggestions. As specified above, the bioluminescence aequorin-based calcium reporter is a quantitative method measuring $[\text{Ca}^{2+}]_i$ (nM) using luminometry while the *GCaMP6s* reporter was used as a fluorescent Ca^{2+} indicator for imaging APC activity.

- p. 32, locomotor assay: Please give reference for pySolo.
Reference added.

- p33, Ramsay assay: Please specify the composition of the *Drosophila* saline.
Done. Information is included at first mention on page 28.

- Fig. S2: legend for g) is missing.
Thanks for bringing this to our attention. Legend for g) has been included.

Reviewer #3 (Remarks to the Author):

The manuscript of Koyama and colleagues describes the physiological relevance of the capa peptide capa receptor system in *Drosophila*, where it has similar functions as the Neuromedin U system of vertebrates. Most importantly, the authors could show formerly unknown physiological roles of capa receptor signaling in a number of different tissues. Two different arms of the capa receptor based signaling system have been identified, one focusing on the intestine and the uptake of nutrients and the other on regulation of the AKH axis. These two axes work together to control nutrient uptake and energy storage.

The manuscript describes a highly interesting topic and is of very high quality (at least in most of its parts). Nevertheless, some issues need clarification:

The most dramatic phenotype was induced by silencing CapR signaling in the intestine, more precisely in the visceral muscle of the intestine. To obtain a better understanding of this system and the multiple actions of Capa on intestinal physiology, the corresponding Capa responsive systems of the intestine have to be described in more detail. This is especially relevant for the enteroendocrine cells and for the copper cell region. Which cells are exactly Capa responsive and which peptide do they express? What is the exact target in the copper cell region? Does reducing CapaR Expression in these cells show the expected phenotype? It is interesting that the reduction of intestinal motility is lethal after a few days. It would be interesting to know if this reduced motility or other factors induce lethality. Related to an earlier question, what is the factor that reduces uptake of nutrients – the reduced gut motility should not be decisive for this?

We thank the reviewer for bringing up these important points. To address these concerns, we first validated the cellular identity of the CapaR-expressing cells by counterstaining *CapaR>mCD8::GFP* flies with an antibody directed against the EE marker, Prospero, as well as a combination of antisera recognizing some of the most abundant peptide hormones (Tachykinin, DH31, CCHa1, AstC and NPF) expressed by the EEs²³. These data show that all the CapaR⁺ cells are Prospero positive (but all Prospero positive cells are not CapaR⁺) and reveal that they express the gut hormones AstC, CCHa1 and Tachykinin but not DH31 or NPF. These data – which have been included in Fig. S2i and mentioned in the main text (page 7, first paragraph) in the revised manuscript – thus verify the cellular identity of the Capa-responsive cells as EEs and further show which peptides they express.

Secondly, we aimed to further dissect the mechanisms underpinning the reduced nutrient absorption phenotype of the *CapaR* depleted flies. To do this we used a GAL4-driver that is highly specific to the midgut acidic zone (*Labial-Gal4*)¹², and used to knockdown a critical subunit of the acid-producing vacuolar H⁺-ATPase (*Vha55-RNAi*) to selectively impair acid production in the copper-cell region of adult flies. These experiments demonstrate that the *Lab^{ts}>Vha55-RNAi* flies show a complete loss of the acidic zone, and that their excreta contain significantly more undigested nutrients and cause a significant reduction in organismal lifespan. These results suggest that acute loss of CCR acidity dramatically impairs nutrient absorption in adult flies, which in combination with gut hypomotility, explain the reduced nutrient absorption and mortality phenotypes displayed by *how^{ts}>CapaR^{KO}* animals. These data have been included in Fig. S5 and described on page 12, first paragraph of the revised manuscript.

References

- 1 MacMillan, H. A. *et al.* Anti-diuretic activity of a CAPA neuropeptide can compromise *Drosophila* chill tolerance. *J Exp Biol* **221**, jeb185884, doi:10.1242/jeb.185884 (2018).
- 2 Koyama, T. & Mirth, C. K. Correction: Growth-Blocking Peptides As Nutrition-Sensitive Signals for Insulin Secretion and Body Size Regulation. *PLoS Biol* **14**, e1002551, doi:10.1371/journal.pbio.1002551 (2016).
- 3 Scopelliti, A. *et al.* A Neuronal Relay Mediates a Nutrient Responsive Gut/Fat Body Axis Regulating Energy Homeostasis in Adult *Drosophila*. *Cell Metab* **29**, 269-284.e210, doi:10.1016/j.cmet.2018.09.021 (2019).
- 4 Sun, J. *et al.* *Drosophila* FIT is a protein-specific satiety hormone essential for feeding control. *Nat Commun* **8**, 14161, doi:10.1038/ncomms14161 (2017).
- 5 Kean, L. *et al.* Two nitridergic peptides are encoded by the gene capability in *Drosophila melanogaster*. *American journal of physiology. Regulatory, integrative and comparative physiology* **282**, R1297-1307, doi:10.1152/ajpregu.00584.2001 (2002).
- 6 Wegener, C., Reinl, T., Jansch, L. & Predel, R. Direct mass spectrometric peptide profiling and fragmentation of larval peptide hormone release sites in *Drosophila melanogaster* reveals tagma-specific peptide expression and differential processing. *J Neurochem* **96**, 1362-1374 (2006).
- 7 Predel, R. *et al.* Peptidomics of CNS-associated neurohemal systems of adult *Drosophila melanogaster*: a mass spectrometric survey of peptides from individual flies. *J Comp Neurol* **474**, 379-392 (2004).
- 8 Santos, J. G., Pollák, E., Rexer, K. H., Molnár, L. & Wegener, C. Morphology and metamorphosis of the peptidergic Va neurons and the median nerve system of the fruit fly, *Drosophila melanogaster*. *Cell Tissue Res* **326**, 187-199, doi:10.1007/s00441-006-0211-7 (2006).
- 9 Tublitz, N. J. & Evans, P. D. Insect cardioactive peptides: cardioacceleratory peptide (CAP) activity is blocked in vivo and in vitro with a monoclonal antibody. *J Neurosci* **6**, 2451-2456, doi:10.1523/jneurosci.06-08-02451.1986 (1986).
- 10 Cognigni, P., Bailey, A. P. & Miguel-Aliaga, I. Enteric neurons and systemic signals couple nutritional and reproductive status with intestinal homeostasis. *Cell Metab* **13**, 92-104, doi:10.1016/j.cmet.2010.12.010 (2011).
- 11 Overend, G. *et al.* Molecular mechanism and functional significance of acid generation in the *Drosophila* midgut. *Sci Rep* **6**, 27242, doi:10.1038/srep27242 (2016).
- 12 Dubreuil, R. R. Copper cells and stomach acid secretion in the *Drosophila* midgut. *The International Journal of Biochemistry & Cell Biology* **36**, 742-752, doi:<https://doi.org/10.1016/j.biocel.2003.07.004> (2004).
- 13 Li, H., Qi, Y. & Jasper, H. Preventing Age-Related Decline of Gut Compartmentalization Limits Microbiota Dysbiosis and Extends Lifespan. *Cell Host & Microbe* **19**, 240-253, doi:<https://doi.org/10.1016/j.chom.2016.01.008> (2016).
- 14 Glukhova, A. *et al.* Rules of Engagement: GPCRs and G Proteins. *ACS Pharmacol Transl Sci* **1**, 73-83, doi:10.1021/acspstsci.8b00026 (2018).

- 15 Bryantsev, A. L., Baker, P. W., Lovato, T. L., Jaramillo, M. S. & Cripps, R. M. Differential requirements for Myocyte Enhancer Factor-2 during adult myogenesis in *Drosophila*. *Developmental biology* **361**, 191-207, doi:10.1016/j.ydbio.2011.09.031 (2012).
- 16 Nongthomba, U., Pasalodos-Sanchez, S., Clark, S., Clayton, J. D. & Sparrow, J. C. Expression and function of the *Drosophila* ACT88F actin isoform is not restricted to the indirect flight muscles. *J Muscle Res Cell Motil* **22**, 111-119, doi:10.1023/a:1010308326890 (2001).
- 17 Weaver, L. N., Ma, T. & Drummond-Barbosa, D. Analysis of Gal4 Expression Patterns in Adult *Drosophila* Females. *G3 (Bethesda)* **10**, 4147-4158, doi:10.1534/g3.120.401676 (2020).
- 18 Mirth, C. & Shingleton, A. Integrating Body and Organ Size in *Drosophila*: Recent Advances and Outstanding Problems. *Frontiers in Endocrinology* **3**, doi:10.3389/fendo.2012.00049 (2012).
- 19 Koyama, T., Texada, M. J., Halberg, K. A. & Rewitz, K. Metabolism and growth adaptation to environmental conditions in *Drosophila*. *Cell Mol Life Sci*, doi:10.1007/s00018-020-03547-2 (2020).
- 20 Gorokhovatsky, A. Y. *et al.* Fusion of *Aequorea victoria* GFP and aequorin provides their Ca²⁺-induced interaction that results in red shift of GFP absorption and efficient bioluminescence energy transfer. *Biochemical and Biophysical Research Communications* **320**, 703-711, doi:<https://doi.org/10.1016/j.bbrc.2004.06.014> (2004).
- 21 Cabrero, P., Richmond, L., Nitabach, M., Davies, S. A. & Dow, J. A. A biogenic amine and a neuropeptide act identically: tyramine signals through calcium in *Drosophila* tubule stellate cells. *Proc Biol Sci* **280**, 20122943, doi:10.1098/rspb.2012.2943 (2013).
- 22 Feng, Y., Ueda, A. & Wu, C. F. A modified minimal hemolymph-like solution, HL3.1, for physiological recordings at the neuromuscular junctions of normal and mutant *Drosophila* larvae. *J Neurogenet* **18**, 377-402, doi:10.1080/01677060490894522 (2004).
- 23 Veenstra, J. A., Agricola, H. J. & Sellami, A. Regulatory peptides in fruit fly midgut. *Cell and tissue research* **334**, 499-516, doi:10.1007/s00441-008-0708-3 (2008).

Reviewer comments, second round –

Reviewer #1 (Remarks to the Author):

Authors have addressed all my previous comments. Many new data have been included. The revised manuscript has met the publication criterion. I recommend it to be accepted for publication.

Reviewer #2 (Remarks to the Author):

In the revised version, I find all my previous issues well and satisfyingly addressed by the authors. I would like to thank for including extra data and carrying out control experiments, especially Uro>tDH44. I have only minor comments, mostly to the new experiments in response to rev#1. These issues should be readily addressable and do not need new experimentation.

- abstract, line 22: "hormonal signals coordinating these functions are largely unidentified". This appears overstated. We know a lot about these signals especially in mammals, and the recent years has also identified several such signals in *Drosophila* and other invertebrates. I therefore suggest to use the same expression as later on in line 52 "...are incompletely characterized..." which certainly is true and does not diminish the significance of your findings.

- results line 466 ff, M&M line 701 ff.: This is a clever experiment, I would not have expected that this is possible, and there remain technical questions which are not appropriately addressed in M&M:

First: there is no isolated bilateral CC, rather it is fused with the hypocerebral ganglion to a median complex. Please specify how you dissected out the CC/HCG, and how you separated them ("...one half fixed immediately, the other half transplanted...").

Second: where did you place the tissue? thorax/abdomen?, dorsal/ventral?. How did you open the body? Please specify to allow to understand how the experiment was actually done.

- line 482: I understand the importance to stress the similarities between vertebrate and insect systems, yet the term "adipose tissue" for flies seems inappropriate. The fat body has functions beyond the "adipose tissue" and this fact is confounded. Please reword appropriately.

- Fig. 4 K: Individual data points should be given similar to Fig. 4 g, i, p-q.

Reviewer #3 (Remarks to the Author):

The revision of the manuscript by Koyama et al. sufficiently clarified my questions. The additional tests performed have provided the necessary clarification. In my opinion, this also applies to the comments of the other reviewers, which were either answered competently or where it was clearly presented that an experimental follow-up of these topics is currently not possible for various reasons.

Final revisions for Nature Communications manuscript NCOMMS-20-35659A

The authors are delighted that the reviewers have agreed that all points previously raised have been addressed in full. We would like to once again express our gratitude for the reviewers for their helpful and constructive criticisms given, which we believe have greatly improved our paper. Our point-by-point response to the points raised by Reviewer #2 is appended below.

Reviewer #1 (Remarks to the Author):

Authors have addressed all my previous comments. Many new data have been included. The revised manuscript has met the publication criterion. I recommend it to be accepted for publication.

Reviewer #2 (Remarks to the Author):

In the revised version, I find all my previous issues well and satisfyingly addressed by the authors. I would like to thank for including extra data and carrying out control experiments, especially Uro>tDH44. I have only minor comments, mostly to the new experiments in response to rev#1. These issues should be readily addressable and do not need new experimentation.

- abstract, line 22: "hormonal signals coordinating these functions are largely unidentified". This appears overstated. We know a lot about these signals especially in mammals, and the recent years has also identified several such signals in Drosophila and other invertebrates. I therefore suggest to use the same expression as later on in line 52 "..are incompletely characterized...." which certainly is true and does not diminish the significance of your findings.

We have changed the statement in the abstract according to the reviewer's suggestion.

- results line 466 ff, M&M line 701 ff.: This is a clever experiment, I would not have expected that this is possible, and there remain technical questions which are not appropriately addressed in M&M:

First: there is no isolated bilateral CC, rather it is fused with the hypocerebral ganglion to a median complex. Please specify how you dissected out the CC/HCG, and how you separated them ("...one half fixed immediately, the other half transplanted...").

Second: where did you place the tissue? thorax/abdomen?, dorsal/ventral?. How did you open the body? Please specify to allow to understand how the experiment was actually done.

The CC/HCG of individual flies were not separated but was transplanted intact along a small piece of esophagus and proventriculus. The two halves refer to two separate populations of CC-containing tissues dissected out from each genotype. This, along with additional information on the protocol for this experiment, has been provided as suggested. The details requested for this experiment is mentioned on page 29, last paragraph of the revised manuscript.

- line 482: I understand the importance to stress the similarities between vertebrate and insect systems, yet the term "adipose tissue" for flies seems inappropriate. The fat body has functions beyond the "adipose tissue" and this fact is confounded. Please reword appropriately.

We have changed "adipose tissue" to "fat body" as suggested.

- Fig. 4 K: Individual data points should be given similar to Fig. 4 g, i, p-q.

According to the author instructions, only plots with a sample size less than 10 should include all data points. The sample size in 4k is higher than 10. However, for consistency we have included all data points in Fig. 4K as well.

Reviewer #3 (Remarks to the Author):

The revision of the manuscript by Koyama et al. sufficiently clarified my questions. The additional tests performed have provided the necessary clarification. In my opinion, this also applies to the comments of the other reviewers, which were either answered competently or where it was clearly presented that an experimental follow-up of these topics is currently not possible for various reasons.